# Uncertainty-aware machine learning to predict non-cancer human toxicity for the global chemicals market

Kerstin von Borries [1] ✉, Katie V. Beckwith [2], Jonathan M. Goodman [2], Weihsueh A. Chiu [3], Olivier Jolliet [1] & Peter Fantke [1,4,5,6] ✉

Humans are exposed to thousands of chemicals, yet limited toxicity data hinder effective management of their impacts on human health. High-performing machine learning models hold potential for addressing this gap, but their uncharacterized prediction performance across the wider range of chemicals undermines confidence in their results. We develop uncertainty-aware models to predict reproductive/developmental and general non-cancer human toxicity effect doses. Our well-calibrated models provide uncertainty estimates aligned with observed prediction errors and chemical familiarity. We predict toxicity with 95% confidence intervals for >100,000 globally marketed chemicals and identify toxicity and uncertainty hotspots. These results can be applied to inform decisions aimed at reducing potential human health impacts and guide targeted data generation and modeling efforts to reduce prediction uncertainty. Here, we show that enhancing transparency in prediction uncertainty provides key insights for building confidence in toxicity predictions, supporting the sound integration of machine learning-based predictions in chemical assessments.

Every day, humans are exposed to a wide range of chemicals through the air, water, food, and countless consumer products. These chemicals can have substantial, yet still largely unknown, toxicity-related impacts on human health[1]. Safeguarding human health requires a sound scientific basis to assess chemical hazards, establish safe exposure limits, regulate harmful substances, and promote the use of safer alternatives. However, the lack of chemical data for the majority of more than 100,000 marketed chemicals registered worldwide has hampered effective chemical assessment[2,3]. A key metric in these assessments is the toxicological point of departure (POD), the lowest dose on the dose-response curve where adverse effects begin to manifest. Lower PODs indicate more potent chemicals, as they require

a smaller exposure dose to elicit toxic effects. Previous studies derived non-cancer PODs using statistical and probabilistic analysis of numerous in vivo animal studies, providing PODs for a few hundred to several thousand chemicals[4–6]. This leaves most globally marketed chemicals without data and hence without appropriate estimates of their non-cancer toxicity potential to harm human health.

In silico tools, particularly machine learning (ML) models, are promising approaches to fill chemical toxicity data gaps[3,7]. They achieve high predictive performance by identifying patterns from large, complex datasets using flexible algorithms such as support vector machines, random forests, gradient boosting machines, and deep learning approaches. However, several challenges hinder the

[1]Quantitative Sustainability Assessment, Department of Environmental and Resource Engineering, Technical University of Denmark, Bygningstorvet 115, 2800, Kgs. Lyngby, Denmark. [2]Centre for Molecular Informatics, Yusuf Hamied Department of Chemistry, University of Cambridge, Lensfield Road, Cambridge, United Kingdom. [3]Department of Veterinary Physiology and Pharmacology, Texas A&M University, 4466 TAMU, College Station, TX, USA. [4]substitute ApS, Graaspurvevej 55, 2400 Copenhagen, Denmark. [5]Department for Evolutionary Ecology and Environmental Toxicology, Goethe University, 60438 Frankfurt am Main, Germany. [6]Department of Environmental Sciences, College of Agriculture and Environmental Sciences, University of South Africa, Florida, Roodepoort, South Africa. ✉e-mail: kejbo@dtu.dk; peter@substitute.dk

widespread adoption of ML models in chemical assessments. Their performance relies heavily on the quantity and quality of available training data[8] and the chosen algorithms and training features[9]. ML models have limited extrapolative capabilities, restricting their applicability to the domains represented by the training data[10,11]. Their high flexibility can also lead to overfitting, where models fit to noise or erroneous data rather than underlying patterns[12], thereby achieving extraordinarily high prediction accuracy on training data, but significantly lower accuracy on validation data sets. As a result, prediction performance can vary significantly across models and chemicals. However, most ML models lack mechanisms to recognize their own limitations or quantify the uncertainty of their predictions. These limitations undermine confidence in predictions, hindering their integration into decision support tools like chemical risk and health impacts assessments[13,14], chemical substitution studies[15,16], safe and sustainable-by-design approaches[17], and life cycle assessment of products and technologies[18].

To improve confidence in ML predictions, substantial research has focused on understanding and quantifying uncertainty in ML models. ML uncertainty is generally divided into two principal types: aleatoric and epistemic uncertainty[19]. Aleatoric uncertainty refers to data-related uncertainty arising from the inherent randomness and variability in the available training data, for example, the uncertainty in PODs derived from small samples of observed effects with substantial biological and experimental variability[20]. Epistemic uncertainty refers to model-related uncertainty caused by a lack of knowledge, which introduces variability in the model-building process, e.g., due to incomplete training data, choice of training features, and uncertain model parameters[19]. A wide range of uncertainty estimation methods have been developed to date. Some focus on aleatoric uncertainty, such as quantile regression, which provides prediction intervals from empirically estimated quantiles. Others target epistemic uncertainty, for example, ensemble methods that estimate variance across point predictions from a large set of predictors. Individually, these uncertainty estimates can be overconfident as they fail to adequately account for both types of uncertainty. In addition, despite these advancements, uncertainty quantification remains underused in ML model development, particularly outside healthcare and autonomous systems as critical applications[21,22].

In this work, we identified uncertainty-aware ML methods (UAMs) that provide quantified confidence intervals with every prediction and developed models for predicting non-cancer PODs for oral exposure. Here, we separate reproductive/developmental from general non-cancer effects to account for the difference in severity at which human lives are impaired[23]. We aimed to develop UAMs that quantify both aleatoric and epistemic uncertainty, generating robust predictions across a wide range of chemical structures, including unfamiliar or challenging compounds, with confidence intervals that accurately reflect these limitations. To demonstrate the broad applicability of this approach, we used the best-performing models to predict reproductive/developmental and general non-cancer PODs for a large set of >100,000 marketed chemicals, providing initial toxicity estimates for widely used chemicals currently lacking POD data. In addition, we analyzed hotspots of high predicted toxicity and related uncertainty to identify trends across chemical classes and highlight priority classes associated with particularly high non-cancer toxicity potency or low prediction confidence. While prior studies have focused on improving predictive accuracy, few have quantified and validated prediction uncertainty, a key barrier to trust in and uptake of predicted data. This study provides the first comprehensive demonstration of uncertainty-aware toxicity modeling at a chemical space of this scale, enabling transparent predictions that support confidence-building in machine learning–based chemical assessments. Unlike prior studies, which have focused mainly on predictive accuracy, we quantify and validate prediction uncertainty, addressing a key limitation to the uptake of predicted data in decision-making tools.

## Results

### Uncertainty-aware ML prediction performance and calibration

We compared the prediction performance and uncertainty calibration of two uncertainty-aware ML methods (UAMs) based on frequentist conformal prediction (CP) and probabilistic Bayesian neural networks (BNNs) for estimating two human toxicity endpoints, namely reproductive/developmental (rd) and general non-cancer (nc) PODs, using datasets covering 2357 and 1845 chemicals, respectively, and comparing four different sets of molecular descriptors as training features.

We found that the random forest-based CP models achieved higher prediction performance on test chemicals with existing data and were better calibrated to quantify uncertainty for these chemicals in comparison to the BNN models based on 10-fold cross-validation (see Fig. 1 for CP models trained with RDKit descriptors, results for other models are shown in Supplementary Figs. 7–13). Overall, models with lower prediction errors (higher prediction accuracy) showed lower average prediction uncertainty, demonstrating the integral relationship between accurate predictions and confidence in these predictions. This link is not always directly used to derive uncertainty estimates: While CP models construct uncertainty estimates from observed prediction errors using a calibration set of chemicals with available data, BNN models derive uncertainty from the posterior distribution of model weights, with higher uncertainty making prediction errors more likely. Despite using different methods and descriptors, predictions across models were highly correlated, with Pearson[2] values ranging from 0.61 –0.92 for median predictions and 0.17–0.64 for uncertainty estimates (see Supplementary Figs. 14–18). This shows that while models generally produce similar median predictions, their uncertainty estimates can vary more substantially. Still, the mean prediction uncertainty across models was correlated with the variability in median predictions across models ($Pearson^2_{rd} = 0.41$ and $Pearson^2_{nc} = 0.49$, see Supplementary Fig. 19). This suggests that the factors contributing to uncertainty within individual models may also drive differences observed across models.

For both endpoints, CP models outperformed BNN models as shown by lower prediction errors and higher coefficients of determination (Fig. 1a, b). Both methods showed lower root mean squared error (RMSE) for $POD_{rd}$ ($RMSE_{CP} = 0.63$, $RMSE_{BNN} = 0.69$) than for $POD_{nc}$ ($RMSE_{CP} = 0.73$ and $RMSE_{BNN} = 0.84$), while the coefficients of determination were slightly higher for $POD_{nc}$ ($R^2_{CP} = 0.55$, $R^2_{BNN} = 0.40$) compared to $POD_{rd}$ ($R^2_{CP} = 0.41$, $R^2_{BNN} = 0.30$). Thus, model predictions captured a larger fraction of the variance in reported $POD_{nc}$, which also exhibited higher variance (1.17) compared to $POD_{rd}$ (0.69). The lower variance in reported $POD_{rd}$ stems from fewer chemicals exhibiting substantial reproductive/developmental effects compared to the wider range of general non-cancer effects in our training dataset (see Supplementary Fig. 1). However, it is important to note that the training data did not indicate the absence of toxicity, e.g. for reproductive/developmental effects, and as regression models, the models will always predict a quantitative POD value.

The overall distribution of prediction uncertainty for $POD_{rd}$ and $POD_{nc}$ showed median 95% confidence intervals (CIs) spanning 2.2 and 2.7 $\log_{10}$-units (i.e. orders of magnitude) for CP models compared to 2.9–3.7 $\log_{10}$-units for BNN models (Fig. 1c, d). This highlights the greater overall confidence of CP models in their predictions compared to BNN models which aligns with the lower prediction errors observed. For most chemicals, CP models in particular quantified uncertainty that is lower than the general variability observed in the data, where 95% of reported $POD_{rd}$ and $POD_{nc}$ values fell within intervals of 3 and 4 $\log_{10}$-units, respectively. The maximum prediction uncertainty for both models (9.7–9.9 and 8.4–9.4 $\log_{10}$-units for $POD_{rd}$ and $POD_{nc}$, respectively) remained near the total reported POD value range (11.4

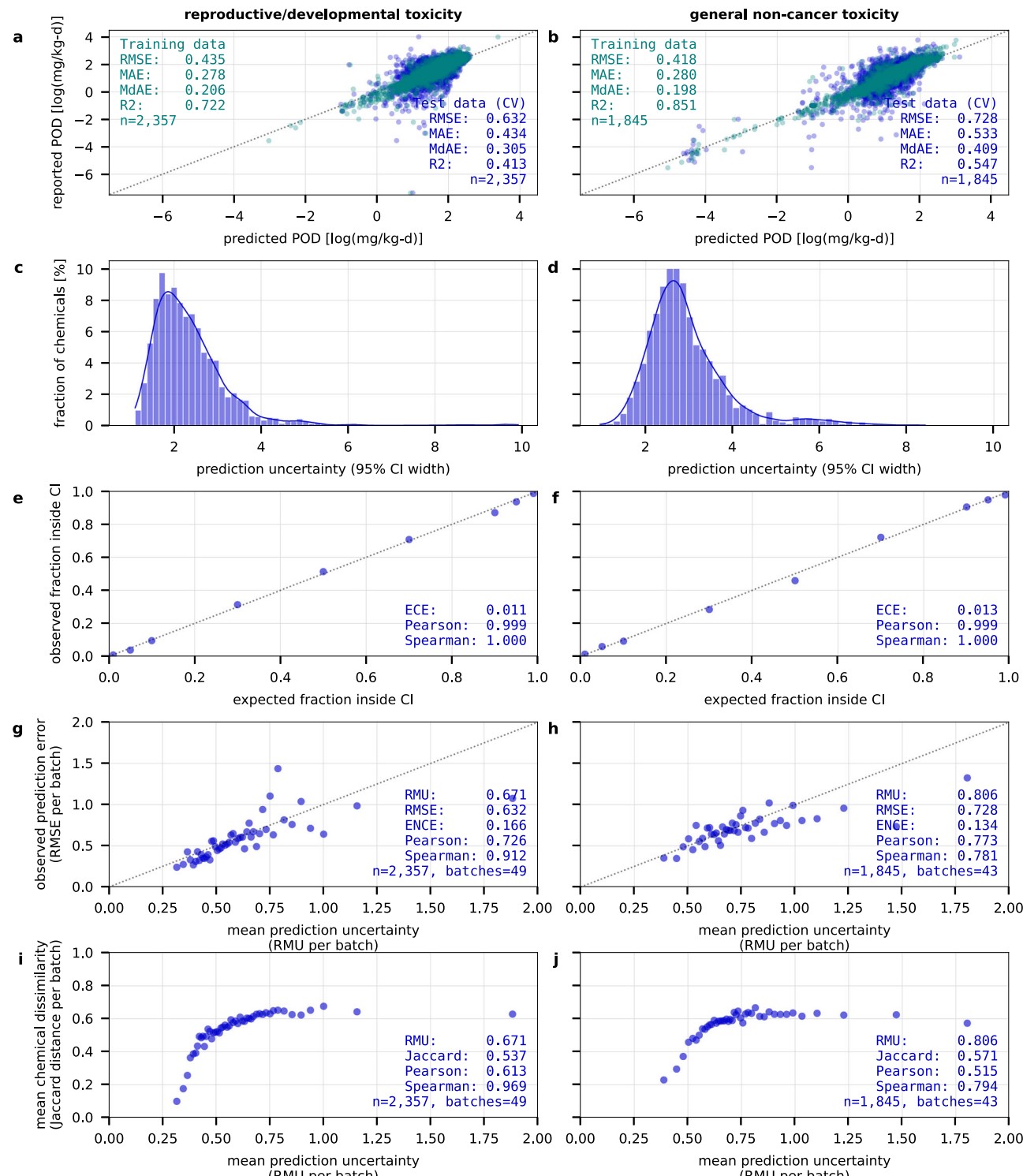

**Fig. 1 | Prediction performance and uncertainty calibration of the CP models. a, b** The model predictions on the training data set (green) and the cross-validated test data set (blue) show a smaller mean prediction error for reproductive/developmental than for general non-cancer toxicity. **c, d** The median prediction uncertainty (95% CI width) spans 2.2 (rd) and 2.7 (nc) $\log_{10}$-units. **e, f** The confidence-based calibration shows an alignment of the expected and observed fractions of measured data points within the predicted CI. **g, h** The error-based calibration shows an alignment of the mean prediction uncertainty (RMU) with the mean prediction error (RMSE) across sorted batches of increasing prediction uncertainty containing 42–49 chemicals each. **i, j** The distance-based calibration shows an

alignment of the mean prediction uncertainty (RMU) with the mean Jaccard distance describing the dissimilarity of a chemical from its five nearest neighbors in the training data set across sorted batches of increasing prediction uncertainty containing 42–49 chemicals each. ECE expected calibration error, ENCE expected normalized calibration error, CI confidence interval, CV cross-validation, Jaccard mean Jaccard distances from 5 nearest neighbors, MAE mean absolute error, MdAE median absolute error, n number of chemicals, Pearson Pearson correlation coefficient, POD point of departure, RMSE root mean squared error, RMU root mean uncertainty, R2 coefficient of determination, Spearman Spearman's rank correlation coefficient. Source data are provided as a Source Data file.

and 9.3 $\log_{10}$-units). The highest and lowest outer uncertainty bounds were within 0.6 $\log_{10}$-units of the highest and lowest reported POD values, indicating that uncertainties are contained within a reasonable range suggested by the data.

The uncertainty quantification provided by both methods was well calibrated to capture different aspects of uncertainty demonstrated by their confidence-based, error-based, and distance-based calibration.

Confidence-based calibration assessed how often the reported value fell within the predicted confidence interval for the test chemicals with available data and whether this matched the expected confidence level, for example, that 95% of the reported points fell within the predicted 95% CI. Both methods showed strong alignment between the expected and observed fractions of reported data points falling within the predicted CI at different confidence levels ranging from 1 to 99% (Fig. 1e, f). The CP models, designed to guarantee a desired confidence level, performed particularly well with an expected calibration error (ECE), describing the mean absolute deviation between expected and observed fractions, of 1.2% (rd, nc). The BNN models showed consistently higher observed fractions than the expected confidence level with an ECE of 5.0% (rd) and 5.1% (nc), indicating slight underconfidence in their predictions, thus consistently overestimating uncertainty.

Error-based calibration assessed whether the models' prediction uncertainty (described by root mean uncertainty, RMU) aligned with the observed prediction error (described by RMSE) across sorted batches of increasing prediction uncertainty containing 42–49 chemicals each. Both methods exhibited the expected trend of increasing RMU with increasing RMSE, although the correlation was less pronounced for BNNs predicting $POD_{nc}$ (Spearman's $\rho$ = 0.44). The expected normalized calibration error (ENCE), describing the mean absolute deviation between the prediction uncertainty and the observed prediction error, was 16.6% (rd) and 13.4% (nc) for CP models compared to 21% (rd) and 20% (nc) for BNN models (Fig. 1g, h). This demonstrated that for both methods, the models' confidence in their predictions reflected their actual prediction accuracy, thus capturing overall trends in prediction errors. The BNN models' trend fell below the identity line, further indicating a more pronounced overestimation of uncertainty.

Distance-based calibration assessed whether higher uncertainty was associated with predicting chemicals that are unlike chemicals seen during training. Both methods showed an increase in chemical dissimilarity (given by the mean Jaccard distance [0,1] to the five nearest training chemicals using Morgan fingerprints) across sorted batches of increasing prediction uncertainty (described by RMU) containing 42–49 chemicals each (Fig. 1i, j). This shows the models' ability to recognize the increase in epistemic uncertainty in regions with little or no precedent data. The shoulder of the trend further suggests that, as a rule of thumb, confident predictions are mainly achievable for chemicals with mean Jaccard distances below 0.5, i.e., chemicals with an average 50% similarity to their five nearest training chemicals.

All in all, while both methods provided well-calibrated uncertainty quantification, the random forest-based CP models were able to fit the observed data better than the BNN models, leading to higher prediction performance and model confidence, thus lower overall uncertainty. This was likely not caused by a lower capacity of BNN's to fit the data, but rather attributable to their weak assumptions requiring larger amounts of data to find a good approximation[19], resulting in larger epistemic uncertainty. In addition, the BNN's performance may have been constrained by limited exploration of network architectures and hyperparameter tuning, settings to which NN performance is more sensitive than random forest-based models. However, increasing network complexity did not consistently improve prediction performance for conventional NNs, supporting the conclusion that data limitations,

rather than model capacity, constrain the NNs ability to learn a better approximation on our datasets (see Supplementary Fig. 5). Based on the model performance and calibration results, we applied the CP models trained on RDKit descriptors in our further analysis.

## Uncertainty-aware ML predictions for more diverse chemicals

Adopting uncertainty-aware ML can be hindered by concerns about sacrificing prediction performance, as these models need to balance reducing prediction errors with providing accurate confidence intervals. We compared the performance of our models with high-performing conventional ML prediction models. For this purpose, we trained five conventional ML algorithms with four types of molecular descriptors and obtained high-performing consensus predictions across these ML models without uncertainty quantification, achieving $RMSE_{rd}$ = 0.59 [0.53, 0.64] and $RMSE_{nc}$ = 0.66 [0.63, 0.69]. As another example, Kvasnicka et al. (2024) achieved similar performance with $RMSE_{rd}$ = 0.58 [0.54, 0.72] and $RMSE_{nc}$ = 0.69 [0.64, 0.76] by applying random forest models, which were trained using predicted physicochemical properties as features, to a comparable set of standardized chemicals with reported PODs[24]. The results indicate that the choice of an uncertainty-aware approach led to a minor decrease in prediction accuracy. For $POD_{rd}$, the mean difference in squared error (0.05) was not statistically significant ($t(9)$ = 1.12, $p$ = 0.26, Cohen's d = 0.03, 95% CI [−0.04, 0.15]), indicating comparable predictive performance. For $POD_{nc}$, the mean difference in squared error (0.09) represented a statistically significant improvement, although effect sizes remained small ($t(9)$ = 2.54, $p$ = 0.01, Cohen's d = 0.08, 95% CI [0.02, 0.16]). At the same time, UAMs enabled us to generate predictions with quantified uncertainty for a larger proportion of chemicals than those typically considered within the model's applicability domain. For instance, based on the threshold of a mean Jaccard distance below 0.5 observed in our results, only 34% of cross-validated chemicals with $POD_{rd}$ and 28% with $POD_{nc}$ would fall within the applicability domain.

To investigate the limits of our UAMs' potential for predicting a more diverse chemical space than applicable for conventional ML models while accurately reflecting varying uncertainty levels, we assessed their performance on a dataset comprising inorganic, metal, and organometallic compounds – chemical classes typically excluded during pre-processing of training datasets due to their non-standard bonding and valencies, which typical molecular features, designed for drug-like chemicals that rarely contain metals, fail to encode. Predictions for these chemicals are thus expected to be highly uncertain. However, if UAMs quantify this uncertainty appropriately, it may reduce the need to exclude non-standardized chemicals during training. This would allow the use of larger, more diverse datasets, further expanding applicability and potentially prediction performance, while also indicating limitations in current training features through uncertainty estimates. We constructed an alternative training dataset that included these non-standardized chemicals, totaling 2873 (rd) and 2225 (nc) compounds. Models were trained using this expanded dataset, incorporating both standardized and non-standardized chemicals, to evaluate their ability to predict uncertainty across a more diverse chemical space. Figure 2a, c show that the predicted $POD_{nc}$ generally aligned with reported POD data for both standardized and non-standardized chemicals, but the models failed to capture a larger fraction of non-standardized chemicals with a reported $POD_{nc}$ below 1 mg/kg-d (22% vs. 14% for standardized chemicals), contributing to higher prediction errors (see Supplementary Fig. 20 for $POD_{rd}$). Fig. 2b illustrates that UAMs addressed the challenge of predicting more diverse data sets by effectively adjusting their prediction uncertainty for standardized chemicals ($RMU_{nc}$=0.82) and non-standardized chemicals ($RMU_{nc}$=1.00), reflecting these differences in observed prediction error. While the prediction uncertainty for standardized chemicals tended to be slightly underconfident compared to the observed error

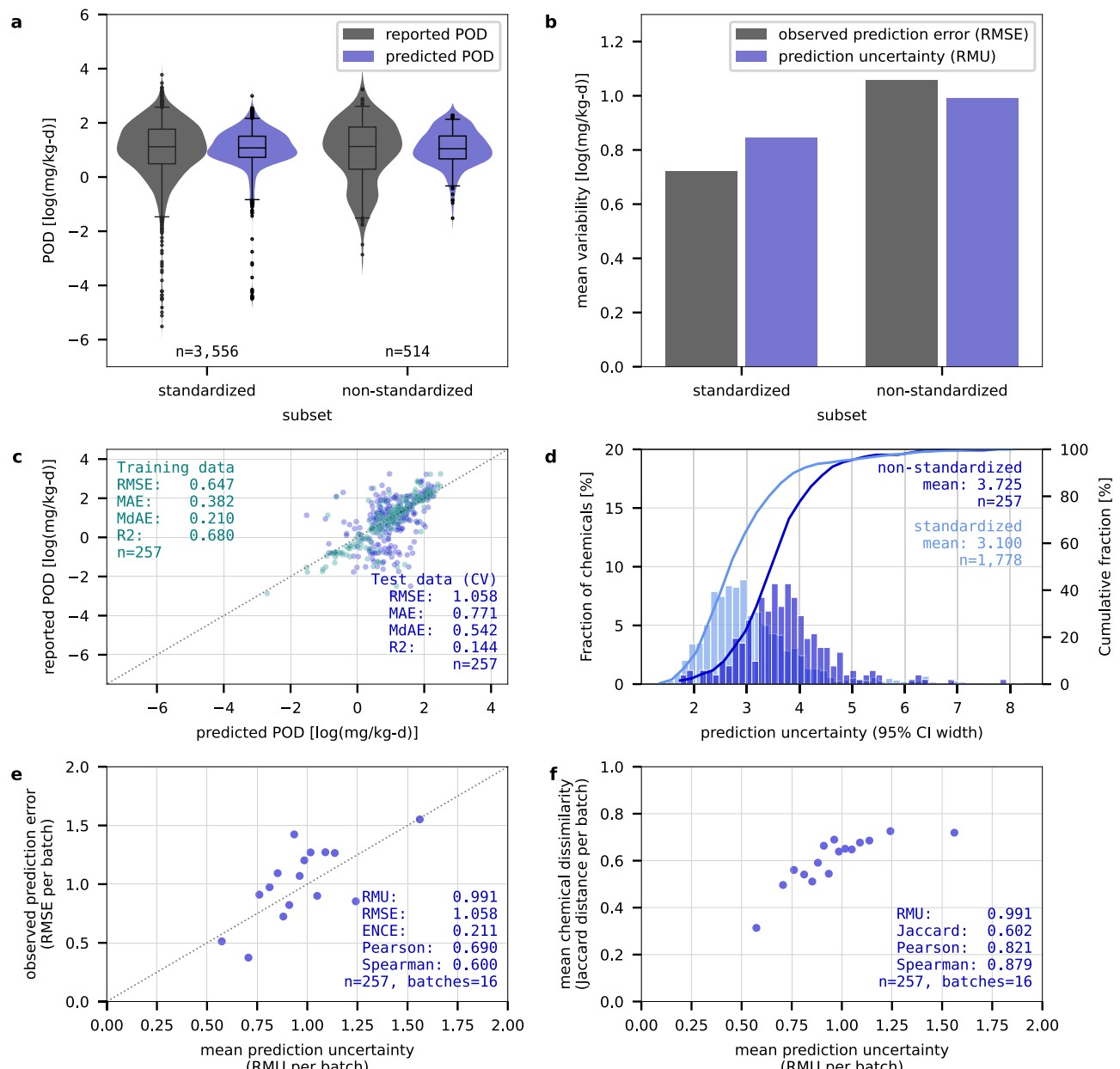

**Fig. 2 | Prediction performance and uncertainty calibration for a non-standardized subset of chemicals.** Predictions are shown for CP models trained with an expanded dataset for POD$_{nc}$. **a** The reported (grey) and predicted (blue) PODs (based on CV) have similar distributions for standardized and non-standardized chemical subsets. **b** The mean prediction uncertainty (RMU, blue) aligns with observed prediction errors (RMSE, grey) for standardized and non-standardized chemical subsets. **c** The model predictions on the training data set (green) and the cross-validated test data set (blue) show higher overfitting for non-standardized chemicals. **d** The prediction uncertainty (95% CI width) spans on average 3.7 log$_{10}$-units for non-standardized chemicals (blue) vs. 3.1 log$_{10}$-units for standardized chemicals (light blue) **e** The error-based calibration for non-standardized chemicals shows an alignment of the mean prediction uncertainty (RMU) with the mean prediction error (RMSE) across batches of increasing

prediction uncertainty. **f** The distance-based calibration for non-standardized chemicals shows an alignment of the mean prediction uncertainty (RMU) with the mean Jaccard distance describing the dissimilarity of a chemical from its five nearest neighbors in the training data set across sorted batches of increasing prediction uncertainty. Boxplots show the median (center) and interquartile range (boxes) with whiskers extending to the 2.5$^{th}$ and 97.5$^{th}$ percentiles. ENCE expected normalized calibration error, CI confidence interval, CP conformal prediction, CV cross-validation, Jaccard mean Jaccard distances from 5 nearest neighbors, MAE mean absolute error, MdAE median absolute error, n number of chemicals, Pearson Pearson correlation coefficient, POD point of departure, RMSE root mean squared error, RMU root mean uncertainty, R2 coefficient of determination, Spearman Spearman correlation coefficient. Source data are provided as a Source Data file.

(RMSE$_{nc}$=0.72), it was slightly overconfident for the non-standardized chemicals (RMSE$_{nc}$=1.04). The error- and distance-based calibration on non-standardized chemicals exhibited a coherent trend in uncertainty estimation (Fig. 2e, f), comparable to what has been observed for the standardized chemicals in Fig. 1. In addition, the prediction

performance for the standardized chemicals was nearly identical to that of models trained on the standardized datasets, demonstrating that training ML models on the expanded dataset allowed them to extract additional information for non-standardized chemicals without compromising standardized chemicals' predictions though mean

prediction uncertainty for standardized chemicals increased slightly from 2.9 to 3.1 log10-units (Fig. 2d). In comparison, when predicting non-standardized chemicals as external test set using the models trained only on standardized datasets, mean prediction uncertainty was 3.9 log10-units as opposed to 3.7 log10-units, despite the models never having observed their limited prediction performance for these chemicals (see Supplementary Figs. 21, 22). In this case, higher prediction uncertainty can be attributed to lower structural similarity between the non-standardized test set and the standardized training set, with a mean Jaccard distance of 0.81 compared to 0.57 among standardized chemicals. This further demonstrates that UAMs can adjust their uncertainty estimates based on both training observations and structural unfamiliarity. However, prediction errors for non-standardized chemicals were substantially higher when predicted with models trained on standardized datasets, leading to overconfident uncertainty estimates and weaker error-based calibration, despite the models increasing their prediction uncertainty. This suggests that uncertainty estimates farther from the training domain should be interpreted as indicative, as the models are forced to extrapolate beyond what they can reliably quantify. It also shows that, although the training features were generally less informative for non-standardized chemicals, they still captured enough structural information, for example, from atom counts, atom fractions, or basic connectivity indices, to enable some degree of predictive learning for these chemicals with the expanded datasets. Nevertheless, prediction uncertainty remained substantially higher for non-standardized chemicals compared to standardized chemicals, limiting the practical utility of these predictions. This highlights that while UAMs can fairly robustly characterize prediction uncertainty for non-standardized chemicals beyond the applicability domain of conventional ML models, tailored strategies are needed for improving prediction performance for these challenging compounds.

## Toxicity and uncertainty hotpots for >100,000 marketed chemicals

Based on our findings we finally applied our CP models trained with the standardized and expanded datasets to predict reproductive/developmental and general non-cancer PODs for a large set of >100,000 globally marketed chemicals[3]. Of these, 126,060 were standardized chemicals, and 8054 were non-standardized chemicals. Results for standardized chemicals are presented here, while provisional results for non-standardized chemicals, marked by high yet potentially underestimated prediction uncertainty, are provided in Supplementary Figs. 24, 25. To spot patterns across marketed chemicals we performed a chemical space analysis following von Borries et al. [3], visualizing predictions in a 2D spatial map using a t-distributed stochastic neighbor embedding (t-SNE)[25], that clusters similar chemical structures closely together (see Fig. 3).

Overall, the median toxicity of our set of marketed chemicals was predicted at 43 mg/kg-d for $POD_{rd}$ and 11 mg/kg-d for $POD_{nc}$ which aligned with the median toxicity in the reported POD data used for training the models. The prediction uncertainty was lower for $POD_{rd}$ with the mean 95% CI spanning 2.6 log10-units compared to 3.1 log10-units for $POD_{nc}$, reflecting the validated prediction performance (2.2 and 2.8 log10-units, respectively). The slight shift towards higher uncertainty can be explained with a shift towards lower chemical similarity relative to the cross-validated test data (see Supplementary Fig. 26). However, similarity remained within a comparable range and did not approach the extreme unfamiliarity observed during external testing with non-standardized chemicals. Therefore, uncertainty estimates are generally expected to align with the calibration established during cross-validation. The prediction uncertainty was distributed asymmetrically between lower and upper bounds, whose distances from the median PODs were on average −1.7 to −1.8 and +1.0 to +1.3 log10-units. Though most marketed chemicals displayed moderate

toxicity with >80% predicted $POD_{rd}$ and $POD_{nc}$ falling between 1 and 100 mg/kg-d, these uncertainties allow differentiating toxicity across chemicals compared to the total variability of 6.5 and 8 log10-units across predicted $POD_{rd}$ and $POD_{nc}$, respectively.

Both endpoints showed several clusters associated with elevated predicted toxicity. Notably, two clusters containing organothiophosphates (Fig. 3a, b, clusters 2 and 3) showed high toxicity with predicted $POD_{rd}$ and $POD_{nc} < 1$ mg/kg-d. Additionally, two large clusters containing per- and polyfluorinated alkyl substances (PFAS) (1) and steroids (8) exhibited elevated toxicity <10 mg/kg-d, in particular for $POD_{nc}$. The clusters containing organothiophosphates and PFAS were also predicted with elevated uncertainty with 95% CIs spanning up to 5 log10-units, while many steroids displayed very high uncertainty with 95% CIs exceeding 6 log10-units (Fig. 3c, d). Organothiophosphates, PFAS, and steroids are thus priority classes both for consideration in chemical assessments aimed at limiting potential harm to human health due to their high predicted toxicity, and as targets for improving model confidence, given their high prediction uncertainty.

Several clusters of polyhalogenated compounds were also predicted to have elevated general non-cancer toxicity (Fig. 3b). These include polybrominated and polychlorinated biphenyls (PBBs, PCBs) and diphenyl ethers (PBDEs, PCBEs) (4, 5), chlorinated benzodioxins and benzofurans (6), and polychlorinated cycloaliphatics (7). The first three clusters showed very high prediction uncertainty, with 95% CIs spanning >6 log10-units (Fig. 3d). These polybrominated and polychlorinated chemical classes are thus also priority classes both for consideration in chemical assessments and as targets for improving model confidence, as they combine high toxicity potency with high prediction uncertainty.

Two additional clusters with notably elevated prediction uncertainty are related to various natural products (9) like alkaloids, polyketides, phenylpropanoids, and lignans, as well as heterocyclic phenothiazines (10). While these natural product classes show only slightly elevated toxicity potency, the substantial prediction uncertainty makes them a priority class for model development efforts aimed at improving model confidence to better assess toxicity potency, particularly given the known biological activity of some chemicals in these clusters like morphinan alkaloids.

## Class-based toxicity ranking for >100,000 marketed chemicals

The chemical space analysis showed that there are substantial trends in both toxicity and uncertainty related to different chemical classes. For a systematic chemical class ranking, we used the ClassyFire chemical taxonomy, grouping chemicals by their lowest available classification level (subclass, class or superclass). Fig. 4 highlights the top 25 chemical classes covering at least 30 marketed chemicals ranked by median predicted toxicity. Table 1 details several top-ranked chemical classes highlighting example chemicals, main uses, and observed adverse effects relevant for humans. The top-ranked chemical classes aligned well with highlighted clusters identified in Fig. 3.

Organothiophosphorus compounds, including thiophosphoric acid esters and dithiophosphate o-esters, ranked high for both toxicity endpoints, covering ~300 chemicals with >40 to 60% in the top 1% of highest overall predicted toxicity. PODs were highest for several globally restricted pesticides, like diazinon, phorate, and ethoprop, and their metabolites, such as phorate oxon, with predicted $POD_{rd}$ below 0.3 mg/kg-d and $POD_{nc}$ below 0.1 mg/kg-d. These chemicals inhibit acetylcholinesterase and other enzymes and form reactive oxygen species, able to induce congenital malformations and fetal death[26–28], as well as neurotoxicity and specific organ toxicities[29,30]. This chemical class was represented in the training data with >20 reported PODs, and showed elevated prediction uncertainty with the median 95% CI spanning 3.3-4.3 log10-units [−1.5-2.0, +1.9-2.1]. Alkyl fluorides, including many PFAS, were ranked with high reproductive/developmental toxicity, covering 444 chemicals with

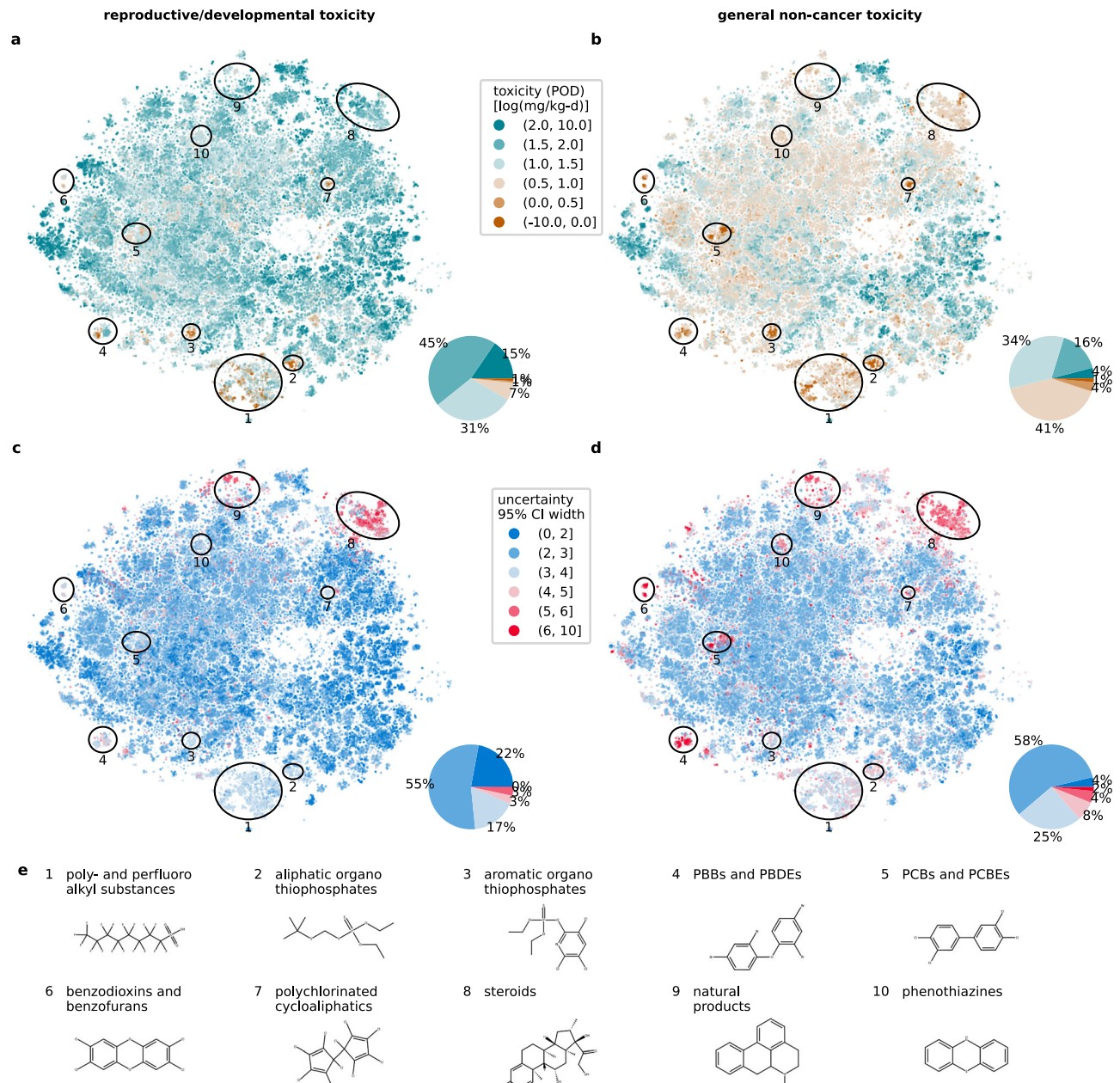

**Fig. 3 | Chemical space maps of 126,060 standardized marketed chemicals.** Maps are colored by **a** reproductive/developmental and **b** general non-cancer toxicity given by the predicted $\log_{10}$-scale POD (green to orange color scale) and **c, d** the associated prediction uncertainty given by the $\log_{10}$-scale width of the 95% CI (blue to red color scale) with **e** illustrative annotations of high toxicity and high uncertainty hotspots. Pie charts illustrate the percentage of marketed chemicals falling into the respective toxicity and uncertainty levels. CI confidence interval, PBB polybrominated biphenyl, PBDE polybrominated diphenylether, PCB polychlorinated biphenyl, PCDE polychlorinated diphenylether, POD point of departure. Source data are provided as a Source Data file.

48.4% in the top 1% of highest predicted toxicity. With a $POD_{rd}$ around 0.15 mg/kg-d, predicted toxicity was highest for perfluorooctanesulfonic acid which has been restricted under the Stockholm Convention since 2009. PFAS have high thermal stability and repel water and dirt, making them attractive in various applications as surfactants, lubricants, and flame retardants. Recent years have, however, seen increasing public concern for the entire group of PFAS, due to their high persistence and bioaccumulation combined with critical health effects, including reduced fertility and offspring health, leading to a proposed EU-wide ban of all PFAS variants covering 10,000 substances[31,32]. This chemical class was represented in the training data with nine reported PODs and showed elevated prediction uncertainty with the median 95% CI spanning 3.2 $\log_{10}$-units [−1.3, +2.0].

The highest-ranking classes for general non-cancer effects were benzo-p-dioxins, covering 96 compounds with 60% in the top 1% of predicted toxicity, and the dioxin-like dibenzofurans, covering 185 compounds with 48% in the top 1% of predicted toxicity levels. Polychlorinated variants like 2,3,7,8-Tetrachlorodibenzodioxin (TCDD) had the highest predicted toxicity ($POD_{nc}$ <$10^{-5}$ mg/kg-d), while non-halogenated variants were predicted as less toxic ($POD_{nc}$ > 10 mg/kg-d). In particular, planar congeners of dioxins and dioxin-like compounds like 2,3,7,8-TCDD show high toxicity by binding to the aryl hydrocarbon receptor (AhR), causing cancer and various non-cancer effects including immunotoxicity, neurological effects, and cardiovascular disease[33]. Despite strict regulation and declining environmental levels, exposure through food from contaminated areas remains a concern[34]. Benzo-p-dioxins and dibenzofurans were

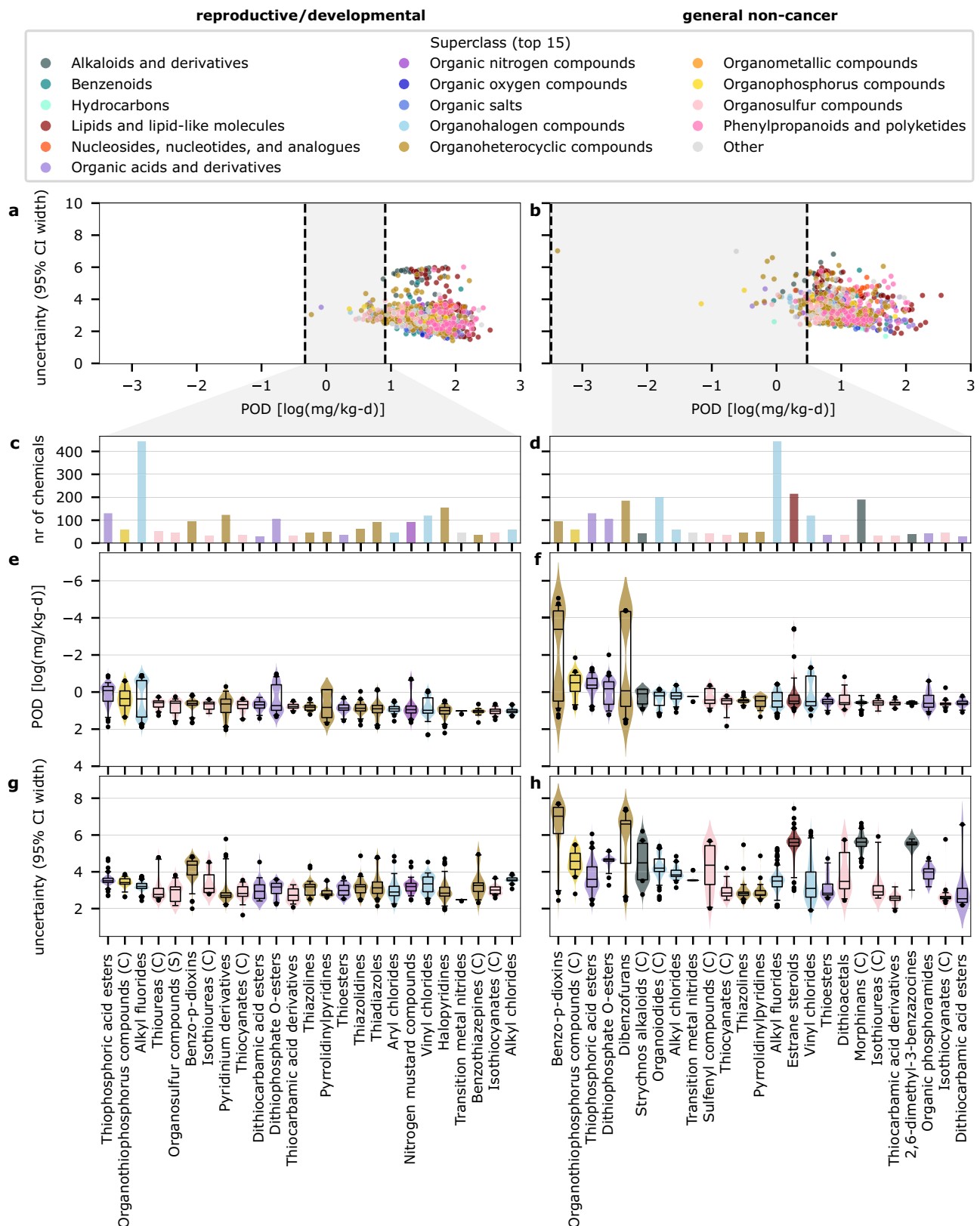

represented by seven and ten reported PODs, respectively. Both classes exhibited substantial variations in predicted toxicity and associated uncertainty, with the 95% CI spanning from 2.4 to 7.7 $\log_{10}$-units. The high uncertainty likely stems from the wide range of toxicity values within the group, driven in part by activity cliffs, where minor structural changes result in significant shifts in toxicity. The highly ranked group of 43 Strychnos alkaloids with a median $POD_{nc}$ around

1 mg/kg-d shows that it is not only synthetic chemicals that are predicted with high toxicity. Strychnines are known neurotoxins that act primarily via inhibiting glycine or acetylcholine receptors, and have been used as pesticides, poisons and in traditional medicine. This chemical class was represented in the training data with only a single reported POD and shows high prediction uncertainty with the median 95% CI spanning 4.5 $\log_{10}$-units [−2.4, +2.0].

**Fig. 4 | Ranking of standardized marketed chemicals grouped by chemical classes.** Chemical classes are based on the lowest classification level (subclass, class (C) or superclass (S)) available in the ClassyFire chemical taxonomy. **a, b** The scatterplot of predicted POD and associated uncertainty given by the 95% CI for all available chemical classes and further highlighting the top 25 chemical classes – ranked by their toxicity potency given by the median predicted POD – covering at least 30 marketed chemicals with **c, d** the number of chemicals in the top 25 chemical classes, **e, f** the distribution of predicted PODs in the top 25 chemical classes and **g, h** the distribution of associated prediction uncertainty given by the 95% CI in the top 25 chemical classes. All data points are colored by their respective superclass among the 15 most frequent ClassyFire superclasses across the set of marketed chemicals (see legend). Boxplots show the median (center) and inter-quartile range (boxes) with whiskers extending to the 2.5th and 97.5th percentiles with underlying sample sizes shown in subfigures (**c, d**). CI confidence interval, POD point of departure. Source data are provided as a Source Data file.

Our results showed that predictions of top-ranked chemical classes generally aligned well with expectations. However, we also observed that the prediction uncertainty tended to be higher for chemicals predicted with higher toxicity. This can be explained with the significantly lower relative representation of high toxicity chemicals in the training data (0.4% with $POD_{rd} < 0.01$ mg/kg-d and 1.5% with $POD_{nc} < 0.01$ mg/kg-d), making high toxicity predictions less likely from the model's perspective and thus decreasing the model's confidence in these predictions. Other factors that may drive uncertainty in individual predictions can be related to insufficient representation of the unique features characterizing a given chemical class or elevated data-related uncertainty in the underlying training dataset. In addition, activity cliffs, where small structural changes result in significant shifts in toxicity can cause very high uncertainty in model predictions - a phenomenon likely responsible for the high uncertainty observed in dioxin-like compounds. This study incorporated additional data on critical dioxin-like chemicals to enhance prediction accuracy; however, boosting model confidence for these chemicals remained challenging. These findings underscore the urgent need to prioritize efforts to predict chemicals with high toxicity and activity cliffs, for instance, by developing models focused on such critical chemical domains that capture subtle differences in molecular structures to enable more confident toxicity estimates. In addition, Fig. 4 shows that the chemical classification used here is not always useful to group chemicals in terms of their toxicity potency. While some classes show very similar behavior in terms of toxicity levels and associated uncertainty, some class definitions seemed to be either too broad or too refined for grouping chemicals in terms of their toxicity potency. For example, organothiophosphates and PFAS were dispersed across multiple chemical classes, while PCBs were part of the broad class of Biphenyls and derivatives containing more than 1500 chemicals. These findings highlight the need for suitable clustering approaches that group chemicals under consideration of their similarity in structure, use and toxic mode of action. While this would be highly useful to translate prediction results into tangible (regulatory) decision support, such data is currently not available for most marketed chemicals.

## Discussion

In this study we applied uncertainty-aware ML to predict reproductive/developmental and general non-cancer PODs with quantified confidence intervals. The approach proved well calibrated to provide uncertainty estimates that align with the expected confidence levels, observed prediction errors and differences in structural similarity with the training chemicals. Applying our models, we provided oral non-cancer toxicity estimates for >100,000 globally marketed chemicals with little formerly available data, covering a large share of the commonly used chemicals in industry and consumer products. These predictions allowed us to identify hotspots in predicted toxicity and related uncertainty across various chemical classes, enabling a broader, group-based perspective that highlighted priority classes to inform decisions limiting the use of potentially hazardous chemicals and guide modeling efforts to improve model confidence for uncertain predictions.

Our models and predicted POD data were developed for application in support of broader chemical impact and risk assessments across marketed chemicals. Such assessments can build on our non-cancer human toxicity predictions as an essential component and combine them with additional information to derive risk or health impact levels, requiring data on chemical production, chemical use in consumer products and industrial processes, indoor and outdoor environmental fate, and human exposure, to quantify chemical-related health risks. For this application, the provided quantitative confidence intervals prevent overconfidence in prediction results and allow verifying if the associated uncertainty can alter or hinder robust conclusions from such chemical assessments. Depending on the decision context, assessments can leverage the best estimates and associated confidence intervals for robust comparative analysis, as in life cycle assessments - or take a more protective approach by considering the upper toxicity bounds to establish safety margins, as is common when assessing human health risks. This is a substantial advancement in departing from the "no data, no problem" mindset toward assessing chemical toxicity potency as needed, based on the most likely or conservative estimates, while leveraging the same data source. In addition, the uncertainty quantification provided can help prioritize experimental studies to reduce uncertainty in challenging chemical domains, iteratively improving the reliability of risk or health impact level assessments. This is an essential step toward harmonizing data generation for chemical risk and sustainability assessments, addressing potentially conflicting results between these two approaches[35] – an increasingly important challenge in the context of integrated frameworks such as the EU Commission's Safe and Sustainable by Design.[17] While implementing such integrated assessments is beyond the scope of the current study, our well-calibrated predictions and confidence intervals provide a solid foundation for future studies to carry out comprehensive risk and impact evaluations.

Further, our results demonstrated how developing UAMs can significantly increase the transparency of prediction uncertainty. Applying this approach to develop models for different parameters, training features, and data sets would help identify the most certain predictions across multiple options, which is a major concern for users of ML models in chemistry applications[36]. This could promote the integration and acceptance of digital prediction methods in chemical assessments, which is currently hindered by low confidence in predicted chemical data. However, enhancing the interpretability of UAMs will be essential for increasing their acceptance, particularly in regulatory contexts. While overall feature importance can be easily obtained (see Supplementary Fig. 23), it is insufficient to interpret predictions in terms of underlying toxicity mechanisms. Combining UAMs with explainable artificial intelligence (XAI) techniques, such as Shapley values, surrogate modeling, or counterfactuals[37], could provide valuable insights into both toxicity mechanisms and uncertainty drivers at the level of individual predictions. However, this integration may face challenges due to the added complexity, computational demands, and technical hurdles of combining both approaches. Additionally, the uncertainty quantified by the UAMs remained high for many predictions, which may limit confidence in their use and highlights the need to further improve prediction accuracy. While uncertainty quantification on its own does not reduce prediction uncertainty, it offers key insights for model developers, helping to identify the factors driving uncertainty and enabling to improve a model's generalizability and reduce prediction uncertainty. Based on insights from the quantified uncertainties, we

**Table 1 | Selection of top 25 chemical classes covering at least 30 out of 126,060 standardized marketed chemicals, ranked by their toxicity potency given by the median predicted reproductive/developmental and general non-cancer points of departure (PODs)**

| Chemical class | nr of chemicals | high potency fraction | CAS | Chemical name | Predicted log10 POD [95% CI] | Reported log10 POD | Chemical class uses and effects |
|---|---|---|---|---|---|---|---|
| **reproductive/developmental effects** | | | | | | | |
| Thiophosphoric acid esters | 130 (10) | 55.4% | 333-41-5 | Diazinon | -0.87 [-1.74, 1.57] | -0.94 | main uses: pesticides; observed effects: female cycle interruption[67], male infertility[68-70], congenital malformation[72,28], fetal death[65-28]; involved mechanisms: inhibition of acetylcholinesterase (AChE) & other enzymes, formation of reactive oxygen species[71], thyroid hormone-receptor binding inhibition, upregulation of estrogen responsive genes[72] |
| | | | 96182-53-5 | Tebupirimfos | -0.87 [-1.74, 1.57] | -1.16 | |
| | | | 90338-20-8 | Butathiofos | -0.87 [-1.77, 1.72] | | |
| Alkyl fluorides | 444 (9) | 48.4% | 1763-23-1 | Perfluorooctane-sulfonic acid | -0.89 [-1.05, 1.64] | -1.29* | main uses: surfactant, lubricant, flame retardant; observed effects: male & female infertility[73,74], altered female cycle[73], lower birth weight[75], offspring male infertility[74,76], offspring immunomodulation[75], offspring neurobehaviour[77]; involved mechanisms: formation of reactive oxygen species[73,78] DNA methylation[73] gene expression[73,77], thyroid hormone disruption[77,78], inhibition of steroidogenic enzymes[74,76], altered nuclear & androgen receptor interactions[74,77,78] |
| | | | 307-35-7 | Perfluorooctane-sulfonyl fluoride | -0.89 [-1.04, 1.62] | -0.91 | |
| | | | 17202-41-4 | Ammonium perfluorononane-sulfonate | -0.85 [-1.00, 1.62] | | |
| Thioureas (C) | 51 (0) | 0 % | 534-13-4 | N,N'-Dimethyl-thiourea | 0.26 [-0.81, 1.66] | | main uses: pesticides, plant-growth regulator, medication[79], metabolite of dithiocarbamates[80]; observed effects: lung damage[81,82], congenital malformation[80,83], female infertility[84]; involved mechanisms: formation of reactive oxygen species[82], complex formation with metalloenzymes[82], thyroid enzyme inhibition[83,84], gene & protein expression[13] |
| | | | 105-55-5 | N,N'-Diethyl-thiourea | 0.44 [-0.67, 2.08] | 0.64* | |
| | | | 62-56-6 | Thiourea | 0.57 [-1.11, 3.66] | 0.74* | |
| **general non-cancer effects** | | | | | | | |
| Benzo-p-dioxins | 96 (7) | 60.4 % | 1746-01-6 | 2,3,7,8-Tetra-chlorodibenzo-p-dioxin | -5.06 [-5.50, 1.44] | -5.52 | main uses: byproducts of industrial and combustion processes[85]; observed effects: immunotoxicity[33], neurological effects[33], liver toxicity[33], diabetes[33], cardiovascular disease[33], skin lesions [33]; involved mechanisms: binding to aryl hydrocarbon receptor (AhR)[33,86], modulated gene expression[33,86], interaction with estrogen receptors[33,86] |
| | | | 50585-46-1 | 1,3,7,8-Tetra-chlorooxanthrene | -4.76 [-5.50, 1.52] | | |
| | | | 262-12-4 | Dibenzo-p-dioxin | 0.95 [-1.05, 2.10] | | |
| Thiophosphoric acid esters | 130 (16) | 75.4 % | 333-41-5 | Diazinon | -1.31 [-2.38, 1.43] | -1.46 | main uses: pesticides; observed effects: neurotoxicity[33,87], neuropsychiatric disorders[30,88], immunotoxicity[30], liver toxicity[29,30], kidney toxicity[30], ocular toxicity[30], metabolic disorder[29], cardiovascular disease[30] and other; involved mechanisms: inhibition of acetylcholinesterase (AChE) and other enzymes[30,87], interaction with acetylcholine receptors[87], formation of reactive oxygen species[30,71] |
| | | | 90338-20-8 | Butathiofos | -1.19 [-2.29, 1.39] | -1.17 | |
| | | | 96182-53-5 | Tebupirimfos | -1.17 [-2.20, 1.28] | -1.17 | |
| Strychnos alkaloids (C) | 43 (1) | | 57-24-9 | Strychnine | -0.15 [-1.76, 1.80] | -0.57 | main uses: pesticide[89], (traditional) medicinal use[89,90], poison[89,90]; observed effects: neurotoxicity (tetanizing, paralyzing)[89,90]; involved mechanisms: inhibition of glycine[89] & acetylcholine receptor[90] |
| | | | 26016-78-4 | 11-Methyl-strychnine | 0.04 [-1.97, 1.97] | | |
| | | | 509-44-4 | 3-Methoxy-strychnidin-10-one | 0.06 [-2.53, 1.98] | | |

Each entry includes chemical class representatives with their CAS registry numbers and chemical names alongside their predicted and reported log10 PODs with 95% confidence intervals (CI) as well as their main uses, observed adverse effects and involved mechanisms in causing these effects. Definitions: nr of chemicals = the total number of marketed chemicals (and chemicals with reported POD used for training) per class, high potency fraction = fraction of chemicals with predicted PODs among the top 1%, * = reported POD was not used for model training (due to <4 underlying data points).

discuss potential strategies to reduce prediction uncertainty in the following. Generally, the epistemic uncertainty of the models can be reduced by increasing the models' knowledge about the prediction task. An essential way to feed knowledge to ML models is through the selection of suitable training features. Commonly used 1D and 2D molecular descriptor approaches, including those utilized in our study, are limited in their ability to fully capture structural features that influence biological interactions and determine toxicity levels. For instance, our approach did not differentiate between enantiomers, which can interact with binding sites differently due to the chirality of most amino acids. Similarly, the spatial arrangement of a molecule, such as the planar structure of 2,3,7,8-TCDD, can significantly affect binding affinity and toxicity, which was not captured by our molecular descriptors. Molecular flexibility, which allows spatial rearrangement to fit various bonding sites, was only partially addressed through the number of rotatable bonds. These limitations could have hindered the models in identifying driving factors behind changes in toxicity, thereby increasing prediction uncertainty. Current descriptor approaches are also limited in handling salts and mixtures, which are commonly addressed by stripping smaller components and retaining only the largest organic component. Around 16% of our set of marketed chemicals were salts or chemical-solvent mixtures that could be affected by an increased uncertainty from this approach. In addition, high, yet underestimated prediction uncertainty was linked to non-standardized chemical classes due to limited applicability of available molecular descriptors, in particular inorganics, metals and organometallics. Developing suitable molecular descriptors that can reflect the behavior of these chemical structures and robustly represent diverse chemical classes and mixtures will be essential to develop accurate and broadly applicable prediction models. Such descriptors could leverage mechanistic understanding to incorporate properties driving chemical activity for these chemical classes, potentially by utilizing intermediate predictions of biologically relevant properties[24] or quantum mechanics-based wave function theory or density functional theory calculations to capture relevant characteristics from an atomic system energy perspective[36]. Alternatively, advanced deep learning-based chemical embeddings could be employed using techniques such as graph neural networks, autoencoders, or natural language processing, as demonstrated by methods like CDDD and mol2vec[38,39]. These approaches offer substantial flexibility in representing chemical structures but are currently limited to small organic molecules and may face challenges in representing more diverse chemicals and mixtures due to the substantially lower data availability. Increasing the models' knowledge can also be achieved by providing more training data, for example through targeted experimental data generation for underrepresented chemicals and target value ranges, particularly those at the extremes of high or low toxicity. In addition, novel data sources like in vitro testing offer lower costs and high-throughput capabilities, which could significantly improve data availability for exposure and effect endpoints[40]. Mechanistically integrating these endpoints into existing chemical assessment frameworks is challenging, as it requires linking them to adverse outcomes at the organism level[41-43]. ML-based approaches could leverage these data to predict conventional endpoints by employing multi-task or transfer learning and imputation approaches which enable models to gain knowledge from related prediction tasks[44,45]. Another way to reduce epistemic uncertainty is by incorporating domain knowledge. For instance, physics-informed ML imposes physical constraints on the prediction task, such as energy and mass conservation laws and reaction kinetics[46], while hybrid modeling combines mechanistic models to represent established knowledge, with ML models to tackle unknown or complex elements such as intricate correlations and high nonlinearity[47-49]. When working to reduce prediction uncertainty, it is important to remember that

aleatoric uncertainty limits how much uncertainty can be minimized. Aurisano et al.[5] quantified substantial aleatoric uncertainty in determining PODs (used for training our models), with a 95% CI spanning >2 $log_{10}$-units due to high variability in effect doses across experiments, species, and effect categories, as well as uncertainties in extrapolating to human equivalent doses. Developing new approaches based on in vitro endpoints with lower aleatoric uncertainty, combined with mechanistic modeling to link them to adverse outcomes at the organism level could potentially achieve lower overall uncertainty[50]. Proper uncertainty quantification will be essential for comparing these approaches to evolving prediction methods building on conventional in vivo data, for which our models can provide a first chemical-dependent benchmark.

The methods outlined in this study are not only applicable for improving non-cancer human toxicity predictions. Future research should focus on extending similar methodologies to other parameters needed to model chemical exposure and effects, including persistence, bioaccumulation, and other types of effects like human cancer toxicity and ecotoxicity. Addressing these data gaps across the broader spectrum of chemicals would allow to more systematically identify chemicals of concern and establish "safe" chemical spaces across marketed and novel chemicals, offering safer alternatives for developing sustainable products and processes.

## Methods

The methodological workflow of this study are outlined in Fig. 5, which provides a schematic overview of the data processing, modeling, and evaluation steps.

### Data set

The toxicity prediction models in this study were built using the extensive datasets of probabilistic PODs derived by Aurisano et al.[5] based on curated in vivo effect test data from the US EPA Toxicity Value Database (ToxValDB). The dataset contained PODs for reproductive/developmental (rd) and general non-cancer effects (nc) via chronic oral exposure for more than 10,000 chemicals that can act as surrogates for PODs derived through authoritative assessment. The separate endpoints allow considering the differing severity between lifelong reproductive/developmental impairments and general non-cancer toxicity effects that typically manifest later in life, when for example used in comparative impact assessment[23]. We complemented the dataset with 26 $POD_{nc}$ for dioxin-like chemicals that were not yet part of the dataset by leveraging toxic equivalency factors (TEFs) reported by DeVito et al. (2024)[51], thereby improving the representation of chemicals with high general non-cancer toxicity. For every chemical in the dataset, chemical structure information in the form of SMILES was collected from the US EPA CompTox Chemicals Dashboard[52] and PubChem[53], which yielded unique chemical structures for 4832 chemicals with $POD_{rd}$ and 5370 chemicals with $POD_{nc}$. Subsequently, both datasets were filtered to keep PODs that were obtained from at least four underlying experimental data points to reduce aleatoric uncertainty as Aurisano et al. derived surrogate PODs using the 25th percentile. The resulting data sets consisted of 2873 chemicals with $POD_{rd}$ and 2225 chemicals with $POD_{nc}$ (expanded dataset). After standardizing chemical structures using the protocol by Mansouri et al. (2018)[54] – which includes the removal of inorganic and organometallic compounds, large molecules (>1000 g/mol), and the stripping of salts and solvents – 2357 unique chemicals with $POD_{rd}$ and 1,945 with $POD_{nc}$ were available for model training and validation (standardized dataset). In addition, the expanded data set was used to assess the potential for achieving robust uncertainty quantification for a more diverse chemical space than applicable for conventional ML models including non-standardized chemicals. The subset of non-standardized chemicals was also used as a challenging external test set for the models trained on standardized chemicals to assess how

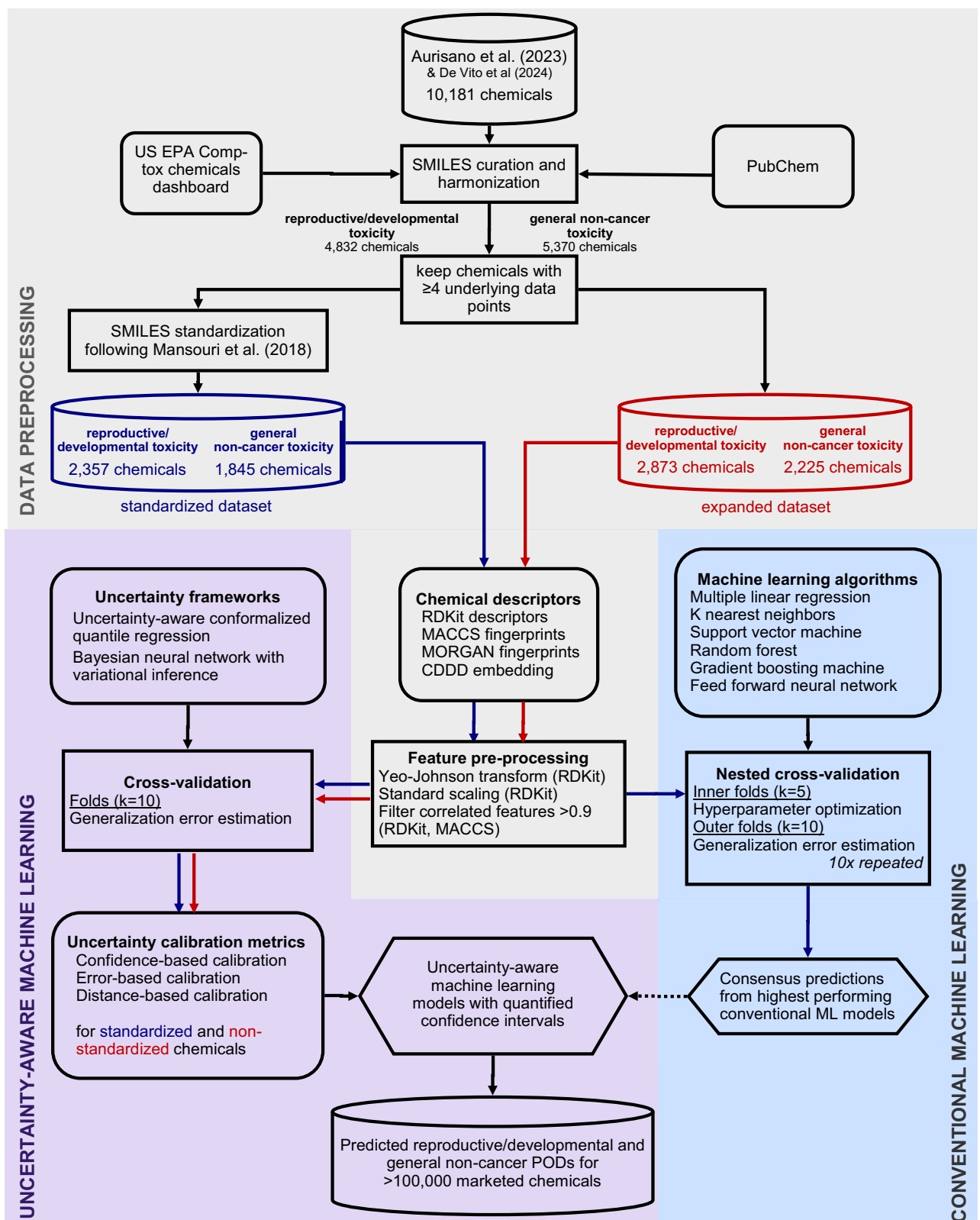

**Fig. 5 | Schematic of the methodological workflow.** The workflow summarizes the data selection and pre-processing, and model training and validation processes for the conventional and uncertainty-aware machine learning models.

uncertainty quantification behaves in entirely unfamiliar chemical domains. All PODs were converted from mg/kg-d to $log_{10}$-transformed units of mol/kg-d prior to training, representing the rational link between molecules and their biological activity[55]. Prediction results were presented in mg/kg-d for easier interpretation.

**Training features**

We considered four types of molecular descriptors that numerically describe chemical structures to be used as training features for the ML models. We chose two types of binary fingerprints (MACCS, Morgan) and two types of continuous descriptors (RDKit, CDDD) (see

Supplementary Table 1 for details). While MACCS and RDKit descriptors are knowledge-based descriptors that encode structural fragments or properties of chemical structures, Morgan fingerprints and CDDD embeddings provide an unweighted representation of a given chemical's full structure. We removed highly correlated features (Pearson correlation coefficient > 0.9) from the knowledge-based descriptors but not from the structural representations as these provide a valid representation only when used together. In addition, we applied a Yeo-Johnson transformation on the RDKit descriptors to achieve Gaussian-like distributions and scaled all RDKit descriptors to zero mean and unit variance. To avoid data leakage in cross-validation, all feature processing parameters were obtained from training sets and applied to respective test sets within each cross-validation fold.

## Conventional ML models without uncertainty quantification

We performed a systematic evaluation of conventional ML algorithms without uncertainty quantification to benchmark performance of best-practice ML model development protocols training each algorithm with the four types of molecular descriptors. For our comparison, we selected six algorithms that cover common algorithm families following different functional principles for finding an optimal fit to a given training data set: Multiple linear regression, k nearest neighbor, support vector machine, random forest, extreme gradient boosting machine and a simple feed forward neural network (see Supplementary section 2.1 for details). For every descriptor-algorithm combination, model performance was assessed through a nested cross-validation procedure with five inner folds to optimize relevant hyperparameters of the respective algorithm and ten outer folds to estimate the generalization error of the model. Chemicals were assigned to each fold using target value-based stratified sampling to compensate impacts from data imbalance. The nested cross-validation was repeated ten times to assess the variation in the generalization error. The generalization error was assessed using mean absolute error (MAE), median absolute error (MdAE), root mean squared error (RMSE) and the (adjusted) coefficient of determination ($R^2$) as performance metrics (see Supplementary Table 4 for details). The assessment allowed us to identify which algorithms, hyperparameter settings and molecular features have high potential to predict PODs from chemical structures. To establish a single, high-performing benchmark for the uncertainty-aware models, we created a consensus model by taking the average across the subset of conventional ML models with the highest statistically significant prediction performances ($p < 0.05$) using pairwise correlated two-sided t-tests based on ten-times repeated cross-validation (degrees of freedom = 9) (see Supplementary section 2.3 for details).

## Uncertainty-aware ML models

To develop uncertainty-aware ML models (UAMs) that provide comprehensive uncertainty quantification covering aleatoric and epistemic uncertainty, we compared two distinct paradigms within ML for quantifying uncertainty, conformal prediction and Bayesian neural networks.

Conformal prediction (CP) is rooted in hypothesis testing following classical, frequentist statistics[19]. It builds confidence intervals around point estimates based on a desired confidence level thereby controlling the fraction of erroneous predictions with a statistically valid probability. CP offers a model-agnostic and non-parametric approach that can be combined with any algorithm and provides confidence intervals without assuming a specific uncertainty distribution. However, not all CP-based methods provide comprehensive uncertainty quantification. We applied uncertainty-aware conformalized quantile regression (UACQR) developed by Rossellini et al. (2023), which addresses both aleatoric and epistemic uncertainty[56]. Rossellini et al. implemented their method with four core algorithms: a quantile linear regression, a light gradient boosting machine with

quantile loss, a quantile regression forest (qRF) and a quantile regression neural network with two hidden layers. In this study, we used qRF as the core algorithm based on our conventional ML model evaluation that showed RF among the high-performing algorithms, and due to its more comprehensive implementation. The CP models were trained with a 95% confidence level.

Bayesian neural networks (BNNs) apply Bayesian principles to model uncertainty and are the main representative for probabilistic modeling approaches[19]. BNNs move beyond deterministic outputs by attaching probability distributions to each model parameter (weights and biases) and returning a probability distribution for the output prediction itself. This allows them to effectively capture both aleatoric and epistemic uncertainty. We defined our BNN with one hidden layer with ReLU (rectified linear unit) activation and an output layer with two nodes assigned to the mean and standard deviation of a Gaussian distribution. The prior distributions of the model parameters were defined as the mixture of two zero-mean normal distributions with different variances and the posterior distributions as multivariate normal distribution following Blundell et al. (2015)[57]. Prediction results were assessed based on random samples of 500 draws from the predicted output distributions.

We assessed the prediction performance and uncertainty calibration of the UAMs through ten-fold cross-validation. Chemicals were assigned to each fold using target value-based stratified sampling to compensate impacts from data imbalance. Due to their unique structures, UAMs were not compatible with conventional hyperparameter optimization tools; hence, reasonable settings were chosen based on experience with equivalent conventional ML models (see Supplementary section 3.2 for details). The overall prediction performance was assessed using the same performance metrics as for the conventional ML models. The quality of the uncertainty quantification was assessed considering three aspects of calibration: confidence-based, error-based, and distance-based calibration. We defined confidence- and error-based calibration following Yang & Li (2023)[58]. The confidence-based calibration compares the expected confidence level and the observed fraction of measured data points falling within the predicted confidence interval. It can be summarized by the expected calibration error (ECE) which describes the mean absolute deviation between expected and observed fractions (see Supplementary Table 6 for details). The error-based calibration curve compares the consistency between the mean observed prediction error and the mean prediction uncertainty across chemical batches of increasing prediction uncertainty. To form the batches, the $n$ testing data points were sorted by the prediction uncertainty and divided into $B$ batches containing $\frac{n}{B}$ data points each. The number of batches was set to the entropy-based optimal number of batches obtained as $\sqrt{n}$[59]. The error-based calibration can be summarized by the expected normalized calibration error (ENCE) defined by Levi et al. (2022)[60] describing the mean absolute deviation between observed error and prediction uncertainty across batches (see Supplementary Table 6 for details). As ENCE has been shown to be sensitive towards the chosen number of batches[59] we assessed the change in ENCE as a function of number of batches ranging from 5 to 200 (see Supplementary Fig. 6). Inspired by the out-of-distribution analysis by Yin et al. (2023)[61], we defined a distance-based calibration to investigate the uncertainty calibration for chemicals that are structurally different from training chemicals. It assesses the consistency between mean structural dissimilarity and mean prediction uncertainty across chemical batches of increasing prediction uncertainty, applying the same batching approach as for error-based calibration. The structural dissimilarity was described by the average Jaccard distance of each test chemical from its five nearest neighbors among training chemicals based on Morgan fingerprints (1024 bits, radius 2). The strength of correlation in all calibration curves was indicated by Pearson and Spearman correlation coefficients (see Supplementary Table 6 for details). The performance of the UAMs

was compared to the consensus of the conventional ML models, and the statistical significance of the difference ($p < 0.05$) was assessed using two-sided student's t-tests based on tenfold cross-validation results from the available dataset of size n (degrees of freedom = $n-1$).

## Model application

We applied the best-calibrated models to predict a set of 134,114 marketed chemicals, which we derived in our previous study[3] from lists of registered chemicals and specific applications provided through US EPA's CompTox Chemicals Dashboard v2.1. Before model application, the models were retrained on all available training data using a 20% split for internal model calibration. Using the standardization protocol by Mansouri et al. (2018)[54], we obtained standardized structures for 126,060 marketed chemicals, which were then predicted using our models trained on the standardized dataset. Preliminary predictions for the remaining 8054 non-standardized marketed chemicals were generated using models trained on the expanded dataset. We visualized the prediction results using t-distributed stochastic neighbor embedding (t-SNE)[25] to map Morgan fingerprints (1024 bits, radius 2) into a two-dimensional chemical space map following von Borries et al. (2023)[3]. All chemicals were assigned to their chemical taxonomy using ClassyFire to enable chemical class-specific interpretation of results[62]. Information on chemical class-related main uses, reported effects, and related mechanisms was obtained by screening available publications on Web of Science between March 2024 and January 2025.

## Inclusion and Ethics

This research did not directly target or involve participants from resource-poor settings or underrepresented groups. No specific inclusion or exclusion criteria were applicable.

## Reporting summary

Further information on research design is available in the Nature Portfolio Reporting Summary linked to this article.

# Data availability

In this work, we relied on previously published and publicly available datasets. To train and validate out machine learning models we used toxicity data reported by ref. 5 and ref. 51. We collected SMILES representations from the US EPA CompTox Chemicals Dashboard (version 2.1, https://comptox.epa.gov/dashboard/, ref. 52,63) and PubChem (https://pubchem.ncbi.nlm.nih.gov/, ref. 53). Chemical classifications were obtained with the web-based application ClassyFire (http://classyfire.wishartlab.com/, ref. 62). All data used and generated in this study are available in the GitHub repository PODUAM (v1.0.0) [https://github.com/kejbo/PODUAM] under accession code https://doi.org/10.5281/zenodo.17951181[64]. The predictions for the large set of marketed chemicals generated in this study are also provided as a structured Excel file in the Supplementary Information. Source data are provided with this paper.

# Code availability

All modeling, data analysis and visualization were performed with custom code developed in Python 3.11 leveraging publicly available python packages. We used RDKit (version 2022.9.5, https://www.rdkit.org/, ref. 65) to calculate RDKit descriptor, MACCS keys and Morgan fingerprints, and the python package CDDD (version 1.0, https://github.com/jrwnter/cddd, ref. 38) to calculate CDDD embeddings. Conventional machine learning algorithms and training support functions were implemented with scikit-learn (version 1.2.2, https://scikit-learn.org/) and TensorFlow (version 2.13.0, https://www.tensorflow.org/). Uncertainty-aware machine learning algorithms were implemented with the python package UACQR (version 2023-06-08, https://github.com/rrross/uacqr, ref. 56) and TensorFlow Probability (version 0.21.0, https://www.tensorflow.org/probability,

ref. 66). Numpy (v1.23.5), Pandas (v1.5.3) and Scipy (v1.10.1) were used for data processing and statistical analysis, and Matpotlib (v3.7.0) and Seaborn (0.13.0) for data visualization. All code generated in this study is available in the GitHub repository PODUAM (v1.0.0) [https://github.com/kejbo/PODUAM] under accession code https://doi.org/10.5281/zenodo.17951181[64]. The models can also be accessed and applied via a Shiny web app at https://dtu-quantitative-sustainability-assessment.shinyapps.io/poduam.

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

## Acknowledgements

The authors would like to thank Prof. Ulf Norinder for sharing his valuable experience with conformal prediction models and Dr. Daniel Yoo for his insightful comments on structuring the manuscript. This work was financially supported by the Swedish Foundation for Strategic Environmental Research (MISTRA) grant DIA 2018/11 (K.B., O.J., P.F.). This work was supported in part by the Engineering and Physical Sciences Research Council (EPSRC) grant EP/S024220/1 (K.V.B.). This work was supported in part by the US National Institutes of Health (NIH) grants P30 ESO29067 and P42 ESO27704 (W.A.C.).

## Author contributions

K.B. contributed to designing the research concept and methodology, model development, data visualization and interpretation, and the original draft of the manuscript. K.V.B. contributed to designing the research methodology, model development, and editing the manuscript. J.M.G. contributed to designing the research methodology, data interpretation, and editing the manuscript. W.A.C. contributed to data interpretation and edited the manuscript. O.J. contributed to data interpretation and edited the manuscript. P.F. contributed to designing the research concept, data interpretation, and editing the manuscript.

## Competing interests

The authors declare no competing interests.
