## [Transparent Peer Review file · Nature Communications]

Uncertainty-aware machine learning to predict non-cancer human toxicity for the global chemicals market

Corresponding Author: Dr Kerstin von Borries

Version 0:

Reviewer comments:

Reviewer #1

(Remarks to the Author)

The manuscript proceeds along previous studies by this team, addressing the prioritisation and screening of the substances and their impact. This topic is quite important. The attention of this manuscript is on the uncertainty when many substances are addressed. Indeed, for the majority of them the experimental data are missing. The manuscript covers two main kinds of uncertainty.

Authors should address two aspects more deeply. (1) the accuracy of the predictions for the substances where data exist and how this can be used to assess uncertainty. (2) the use of different predictive systems, since the predictions can vary depending on them.

(Remarks on code availability)

Reviewer #2

(Remarks to the Author)

The authors developed uncertainty aware models that predict reproductive and non-cancer human toxicity of more than 130K marketed chemicals. The major contributions of this work are presented as 1) reducing human health risks and 2) improving model performance. These contributions appear to be somewhat overstated, as the authors do not present evidence that this work has reduced human health risk (though application of the knowledge produced by the authors certainly has the potential to be applied in such a way that could augment standard practices for identifying potentially toxic chemicals and thereby reduce human health risk), and the generation of 95% confidence intervals in estimating aleatoric and epistemic uncertainty for conformal and Bayesian network-generated predictions lacks novelty. Readers may reasonably interpret results as contributing primarily to a relatively new prediction task that estimates the risk for reproductive and non-cancer human toxicity of chemicals, some of which may not have been previously assessed for toxicity by quantitative methods, which is a considerable contribution, but would seem more appropriate for a more targeted audience.

(Remarks on code availability)

Reviewer #3

(Remarks to the Author)

The study addresses the significant lack of toxicity information for most marketed chemicals by developing uncertainty-aware machine (UAM) learning models. For over 130,000 chemicals, these models predict toxicological points of departure (PODs) for general non-cancer harm as well as reproductive/developmental impacts on a broad scale. The study also identifies chemical groups posing high toxicity risks and areas where prediction confidence is low. In addition, the manuscript is well-organized, and figure 5 provided a very clear overview of the methodological workflow. The statistical analysis is comprehensive and in-depth and the topic will have high impact. While acknowledging these important work, I have a few minor points for consideration,

1) Although conformal prediction (CP) seems perform better than Bayesian neural networks (BNNs), the performance of BNN may be limited by the network structure, training strategy (the range of candidate hyper-parameters), or data volume.

2) While developed uncertainty-aware models (UAMs) measure the uncertainty and quantify prediction confidence, the random forests or neural networks are still "black box" models. The predictions are based on correlations in the data and may not fully capture the underlying mechanisms of toxicity. Enhancing the interpretability of the model to help understand the driving factors behind the predictions is important for regulatory applications.

3) The validation for "non-standardized" chemicals are based on training/testing within an expanded dataset. This might not fully represent performance on truly unseen, challenging chemical structures compared to using a dedicated external set of such compounds. If it is feasible, the validation on a truly external dataset would support claims about model generalizability and robustness.

4) The generation of this extensive prediction dataset with uncertainty and PODs hold significant potential value. Are there any plans to make these predictions and their uncertainties known to the wider scientific community or fully public accessible?

(Remarks on code availability)

The overall quality of the repository and its reproducibility is very good. Several cross-validation scripts have been tested and yielded similar results with the reported findings. Here are some suggestions for implementation and replication,

- The Mac OS users need to install CMake to compile code for some dependencies (e.g. dm-tree). The The README should add instructions on install build tools.

- The README doc lacks instructions and commands on how to run the experiments. Adding sample commands or scripts would improve readability. For example, adding the command `$ python ./manuscript/2-1_performance/run_cv_BNN.py` or `$ python train_final_models.py` would be helpful.

- For all training files, the replication could be improved by avoiding hardcoding values for parameter settings. Authors could consider use package like argparse to allow for more convenient replication or experiments, or maybe consider use configuration file (e.g., JSON) for easily editing the key parameters.

- The current scripts assume that hyper-parameters are already considered "optimal". The nested cross-validation approach with hyper-parameters optimization has been described in the supplementary section but not mentioned in the repository. The README could provide a navigation to that supplementary section for further implementation details.

Version 1:

Reviewer comments:

Reviewer #1

(Remarks to the Author)

The revised manuscript clarified the opne points. The manuscript is well-structured and explicative.

(Remarks on code availability)

Reviewer #2

(Remarks to the Author)

The revised manuscript now describes the potential to apply results from the manuscript to inform decisions regarding limitations on use of potentially harmful chemicals and the potential to guide future improvements in predictive performance for chemical classes that are difficult to predict due to high uncertainty. These revisions incompletely address concerns regarding relatively weak novelty and the important potentials described in the revision – though testable – remain theoretical at this stage of investigation. This is important work that when carried forward to realize the above potentials will represent a major scientific contribution.

(Remarks on code availability)

Reviewer #3

(Remarks to the Author)

The authors have addressed all my concerns. I appreciate their thorough response and efforts to improve the manuscript. In addition, the code has been significantly improved in clarity and documentation. I recommend this manuscript for publication.

(Remarks on code availability)

Detailed answers to reviewers' suggestions:

Reviewer #1

General comment: *The manuscript proceeds along previous studies by this team, addressing the prioritisation and screening of the substances and their impact. This topic is quite important. The attention of this manuscript is on the uncertainty when many substances are addressed. Indeed, for the majority of them the experimental data are missing. The manuscript covers two main kinds of uncertainty. Authors should address two aspects more deeply. (1) the accuracy of the predictions for the substances where data exist and how this can be used to assess uncertainty. (2) the use of different predictive systems, since the predictions can vary depending on them.*

Answer and action: Thank you for the thoughtful review of our manuscript. The accuracy of predictions for substances with available data is shown in Figure 1 for both the training and cross-validated test chemicals. In the manuscript, we have generally used the term “prediction error” rather than “prediction accuracy”, as the latter is more commonly associated with classification models. However, we recognize that “accuracy” may be more intuitive for some readers, so we have clarified this terminology in the revised text. We also added an explanation of the relationship between prediction error (or “accuracy”) and prediction uncertainty. Finally, we included an additional analysis to examine how strongly predictions and uncertainty estimates correlate across the different predictive systems used in this study. The relevant changes are presented below:

L136: *“We found that the random forest-based CP models achieved higher prediction performance on test chemicals with existing data and were better calibrated to quantify uncertainty for these chemicals in comparison to the BNN models based on 10-fold cross-validation (see Figure 1 for CP models trained with RDKit descriptors, results for other models are shown in SI, Figures S1-7 to S1-13). Overall, models with lower prediction errors (higher prediction accuracy) showed lower average uncertainty, demonstrating the integral relationship between accurate predictions and confidence in these predictions. This link is not always directly used to derive uncertainty estimates: While CP models construct uncertainty estimates directly from observed prediction errors using a calibration set of chemicals with available data, BNN models' uncertainty estimates derive from uncertainty in model weights, with higher uncertainty making prediction errors more likely.”*

L146: *“Despite using different methods and descriptors, predictions across models were highly correlated, with Pearson² values ranging from 0.61 to 0.92 for median predictions and 0.17 to 0.64 for uncertainty estimates (see SI, Figures S1-14 to S1-18). This shows that while models generally produce similar median predictions, their uncertainty estimates can vary more substantially. Still, the mean prediction uncertainty across models was correlated with the variability in median predictions across models (Pearson²_{rd}=0.41 and Pearson²_{nc}=0.49, see SI, Figure S1-19). This suggests that the factors contributing to uncertainty within individual models may also drive differences observed across models.”*

The following key figures have been added to the supplemental file S1:

- Figure S1-14 presents heatmaps showing the correlation between prediction medians and uncertainty estimates across prediction systems (detailed scatter plots underlying the correlation values are presented in Figures S1-15 to S1-18),
- Figure S1-19 shows how the variability in median predictions across model systems correlates with the mean prediction uncertainty.

Figure S1-14. Correlation heatmaps of median predictions and uncertainty estimates (given by 95% CI width) for reproductive/developmental (left) and general non-cancer (right) PODs across different methods and descriptors given by Pearson² values. **Abbreviations:** BNN = Bayesian neural network, CDDD = CDDD embedding, CI = Confidence interval, CP = Conformal prediction, MACCS = MACCS keys, Morgan = Morgan fingerprints, RDKit = RDKit descriptors

Figure S1-19. Correlation scatter plots with $Pearson^2$ for variability (given as 95% CI width) across median predictions by different models and the prediction uncertainty (given by 95% CI width) averaged across different models for reproductive/developmental and general non-cancer PODs.

Reviewer #2

Comment: *The authors developed uncertainty aware models that predict reproductive and non-cancer human toxicity of more than 130K marketed chemicals. The major contributions of this work are presented as 1) reducing human health risks and 2) improving model performance. These contributions appear to be somewhat overstated, as the authors do not present evidence that this work has reduced human health risk (though application of the knowledge produced by the authors certainly has the potential to be applied in such a way that could augment standard practices for identifying potentially toxic chemicals and thereby reduce human health risk), and the generation of 95% confidence intervals in estimating aleatoric and epistemic uncertainty for conformal and Bayesian network-generated predictions lacks novelty. Readers may reasonably interpret results as contributing primarily to a relatively new prediction task that estimates the risk for reproductive and non-cancer human toxicity of chemicals, some of which may not have been previously assessed for toxicity by quantitative methods, which is a considerable contribution, but would seem more appropriate for a more targeted audience.*

Answer and action: Thank you for your thoughtful review and for highlighting your concerns regarding our manuscript. We appreciate the opportunity to clarify our contribution and have revised the text accordingly.

Our main objective has been to help close the substantial data gap on non-cancer human toxicity effects by developing predictive models and generating data with quantified uncertainty for globally marketed chemicals. The purpose of developing uncertainty-aware ML models with quantified uncertainty was not to improve model

performance per se, but rather make prediction performance across chemicals more transparent. While we agree that the application of Conformal Prediction and Bayesian approaches is not novel in itself, they are still not widely adopted in cheminformatics or many other ML application areas. We believe that our application-focused evaluation of these methods provides value both to developers of such algorithms and to domain experts building or using chemical prediction models, which is one reason why we reach for a broader audience.

As pointed out in your review, by identifying chemical classes with high predicted toxicity and/or uncertainty, our results can be *applied* to: a) inform decisions to limit the use of potentially harmful chemicals without toxicity estimates based on experimental data, and b) guide efforts to improve model predictions for chemical classes with high prediction uncertainty – an opportunity for which we discuss different targeted data generation and modelling strategies.

We have revised the manuscript to further clarify this distinction between main contributions and intended applications and to prevent any potential misunderstanding.

Abstract

L39: “These *results can be applied to inform decisions aimed at reducing potential human health impacts and guide data generation and modelling efforts to reduce prediction uncertainty.*”

Highlights

L63: “*Identified hotspots of human toxicity and uncertainty can inform decisions to limit the use of potentially hazardous chemicals as well as guide future targeted data generation and modelling efforts to reduce uncertainty in predicted toxicity data.*”

Introduction

L99: “*To improve confidence in ML predictions, substantial research has focused on understanding and quantifying uncertainty in ML models. ML uncertainty is generally divided into two principal types: aleatoric and epistemic uncertainty²². Aleatoric uncertainty refers to data-related uncertainty arising from the inherent randomness and variability in the available training data, for example, the uncertainty in PODs derived from small samples of observed effects with substantial biological and experimental variability²³. Epistemic uncertainty refers to model-related uncertainty caused by a lack of knowledge, which introduces variability in the model-building process, e.g., due to incomplete training data, choice of training features, and uncertain model parameters²². A wide range of uncertainty estimation methods have been developed to date. Some focus on aleatoric uncertainty, such as quantile regression, which provides prediction intervals from empirically estimated quantiles. Others target epistemic uncertainty, for example, ensemble methods that estimate variance across point predictions from a large set of predictors. Individually, these uncertainty estimates tend to be overconfident as they fail to adequately account for both types of uncertainty. In addition, despite these advancements, uncertainty quantification remains underused in ML model development, particularly outside healthcare and autonomous systems as critical applications^{19,20}.*”

L125: “*In addition, we analyzed hotspots of high toxicity and related uncertainty to identify trends across chemical classes and highlight priority classes associated with particularly high non-cancer toxicity potency or low prediction confidence.*”

Results

L375: “*Organothiophosphates, PFAS, and steroids are thus priority classes both for*

consideration in chemical assessments aimed at limiting potential harm to human health due to their high predicted toxicity, and as targets for improving model confidence, given their high prediction uncertainty.”

L383: *“These polybrominated and polychlorinated chemical classes are thus also priority classes for consideration in chemical assessment and as targets for improving model confidence as they combine high toxicity potency with high prediction uncertainty.”*

L389: *“While these natural product classes show only slightly elevated toxicity potency, the substantial prediction uncertainty makes them a priority class for model development efforts aimed at improving model confidence to better assess toxicity potency, particularly given the known biological activity of some chemicals in these clusters like morphinan alkaloids.”*

Discussion

L517: *“Our models and predicted POD data were developed for application in support of broader chemical impact and risk assessments across marketed chemicals by combining our non-cancer human toxicity predictions with exposure estimates from experimental data or exposure models.”*

L547: *“Though uncertainty quantification on its own does not reduce prediction uncertainty, it offers key insights for model developers, helping to identify the factors driving uncertainty and enabling to improve a model’s generalizability and reduce prediction uncertainty.”*

Reviewer #3

General comment: *The study addresses the significant lack of toxicity information for most marketed chemicals by developing uncertainty-aware machine (UAM) learning models. For over 130,000 chemicals, these models predict toxicological points of departure (PODs) for general non-cancer harm as well as reproductive/developmental impacts on a broad scale. The study also identifies chemical groups posing high toxicity risks and areas where prediction confidence is low. In addition, the manuscript is well-organized, and figure 5 provided a very clear overview of the methodological workflow. The statistical analysis is comprehensive and in-depth and the topic will have high impact. While acknowledging these important work, I have a few minor points for consideration,*

Answer: Thank you for the thoughtful review of our manuscript and the many valuable suggestions. We have carefully considered each of your comments and provide detailed point-by-point responses below.

Comment 1: *Although conformal prediction (CP) seems [to] perform better than Bayesian neural networks (BNNs), the performance of BNN may be limited by the network structure, training strategy (the range of candidate hyper-parameters), or data volume.*

Answer and action: We agree that BNNs are very powerful and could perform equally well, if not better than, CP models. Since increasing network complexity combined with more extensive hyperparameter tuning to conventional NNs did not substantially improve prediction performance on our data set, we concluded that the performance of both NNs and BNNs in this work is primarily constrained by data availability. We expanded this

explanation in the manuscript and included a supplementary figure (S1) illustrating prediction performance across wider and deeper architectures.

L217: “This was likely not caused by a lower capacity of BNN’s to fit the data, but rather attributable to their weak assumptions requiring larger amounts of data to find a good approximation, resulting in larger epistemic uncertainty. In addition, the BNN’s performance may have been constrained by limited exploration of network architectures and hyperparameter tuning, settings to which NN performance is more sensitive than random-forest based models. However, increasing network complexity combined with hyperparameter tuning did not consistently improve prediction performance for conventional NNs, supporting the conclusion that data limitations, rather than model capacity, constrain the BNNs ability to learn a better approximation on our datasets (see SI, Figure S1-5).”

Figure S1-5. Changes in prediction error (given by RMSE) for different neural network architectures in terms of number and size of hidden layers. Hidden layer sizes are defined relative to the input size (p) as multiples of p . The figure shows that changes towards deeper and wider architectures did not lead to substantial reductions in prediction error, in particular for cross-validated test set performance. Boxplots show the interquartile range (boxes) with whiskers extending to the 2.5th and 97.5th percentiles.

Comment 2: While developed uncertainty-aware models (UAMs) measure the uncertainty and quantify prediction confidence, the random forests or neural networks are still “black box” models. The predictions are based on correlations in the data and may not fully capture the underlying mechanisms of toxicity. Enhancing the interpretability of the model to help understand the driving factors behind the predictions is important for regulatory applications.

Answer and action: We fully agree that improving interpretability of ML-based

predictions is a critical challenge, particularly for regulatory applications. This challenge becomes even greater when developing models intended to cover a broad chemical space, as toxicity mechanisms can differ substantially across domains. Similar to how uncertainty-aware ML shifts from defining a global applicability domain to estimating uncertainty for individual predictions, mechanistic interpretation will also need to occur at the level of individual predictions, which can be addressed by explainable AI (XAI) methods. Combining uncertainty-aware ML with XAI is an important future direction but falls beyond the scope of this manuscript. However, we have included general feature importance of our models in supplemental figure S1- 23 and expanded the discussion to acknowledge the need for future research in this area.

L539: *“In addition, enhancing the interpretability of UAMs will be essential for increasing their acceptance, particularly in regulatory contexts. While overall feature importance can be easily obtained (see SI, Figure S1-23), it is insufficient to interpret predictions in terms of underlying toxicity mechanisms. Combining UAMs with explainable artificial intelligence (XAI) techniques, such as Shapley values, surrogate modelling or counterfactuals⁶¹, could provide valuable insights into both toxicity mechanisms and uncertainty drivers at the level of individual predictions. However, this integration may face challenges due to the added complexity, computational demands, and technical hurdles of combining both approaches.”*

Figure S1-23. Feature importance distributions from 1,000 quantile regression forests in the CP models trained with the standardized datasets, shown for the 50 most important features ranked by mean importance for PODrd (left) and PODnc (right).

Comment 3: The validation for "non-standardized" chemicals are based on training/testing within an expanded dataset. This might not fully represent performance on truly unseen, challenging chemical structures compared to using a dedicated external set of such compounds. If it is feasible, the validation on a truly external dataset would support claims about model generalizability and robustness.

Answer and action: Thank you for this interesting suggestion. Our primary goal in assessing UAM performance on an expanded training dataset was to evaluate whether these models can learn from a more diverse chemical space than conventional ML models, while accurately reflecting varying levels of uncertainty. This was of interest, as it could reduce the need to exclude non-standardized chemicals from training, enabling larger and more diverse datasets, thereby further expanding prediction applicability. Stress-testing the models by predicting non-standardized chemicals exclusively as an external test set indeed provides additional insight into how effectively uncertainty

estimates can serve as a warning when predictions move far outside the trained domain. We agree this analysis could strengthen the manuscript and have added clarifications on our initial approach and our additional findings as presented below.

L266: *“To investigate the limits of our UAMs’ potential for predicting a more diverse chemical space than applicable for conventional ML models while accurately reflecting varying uncertainty levels, we assessed their performance on a dataset comprising inorganic, metal, and organometallic compounds – chemical classes typically excluded during pre-processing of training datasets due to their non-standard bonding and valencies, which typical molecular features, designed for drug-like chemicals that rarely contain metals, fail to encode. Predictions for these chemicals are thus expected to be highly uncertain. However, if UAMs quantify this uncertainty appropriately, it may reduce the need to exclude non-standardized chemicals during training. This would allow the use of larger, more diverse datasets, further expanding applicability and potentially prediction performance, while also indicating limitations in current training features through uncertainty estimates.”*

L297: *“In comparison, when predicting non-standardized chemicals as external test set using the models trained only on standardized datasets, mean prediction uncertainty was 3.9 log₁₀-units as opposed to 3.7 log₁₀-units, despite the models never having observed their limited prediction performance for these chemicals (see SI Figure S1-21,22). In this case, higher prediction uncertainty can be attributed to lower structural similarity between non-standardized test set and standardized training set, with a mean Jaccard distance of 0.81 compared to 0.57 among standardized chemicals. This further demonstrates that UAMs can adjust their uncertainty estimates based on both training observations and structural unfamiliarity. However, prediction errors for non-standardized chemicals were substantially higher when predicted with models trained on standardized datasets, leading to overconfident uncertainty estimates and weaker error-based calibration, despite the models increasing their prediction uncertainty. This suggests that uncertainty estimates farther from the training domain should be interpreted as indicative, as the models are forced to extrapolate beyond what they can reliably quantify. It also shows that, although the training features were generally less informative for non-standardized chemicals, they still captured enough structural information, such as atom counts, fractions, or basic connectivity indices, to enable some degree of predictive learning for these chemicals with the expanded datasets.”*

L356: *“The slight shift towards higher uncertainty can be explained with a shift towards lower chemical similarity relative to the cross-validated test data (see SI, Figure S1-26). However, similarity remained within a comparable range and did not approach the extreme unfamiliarity observed during external testing with non-standardized chemicals. Therefore, uncertainty estimates are generally expected to align with the calibration established during cross-validation.”*

Figure S1-21. External prediction performance and uncertainty calibration for the non-standardized test set predicted by CP models trained with the standardized dataset for POD_{rd} . **(a)** Prediction performance on the challenging, external test data set of non-standardized chemicals **(b)** The mean prediction uncertainty (RMU) and observed prediction errors (RMSE) for non-standardized chemicals compared to standardized chemicals (during cross-validation). **(c)** Distribution of prediction uncertainty (95% CI width) for non-standardized chemicals compared to standardized chemicals (during cross-validation) **(e)** Confidence-based calibration curve for non-standardized chemicals **(f)** Error-based calibration curve for non-standardized chemicals. **Abbreviations:** ENCE=expected normalized calibration error, CI=confidence interval, CV=cross-validation, Jaccard=mean Jaccard distances from 5 nearest neighbors, MAE=mean absolute error, MdAE=median absolute error, n=number of chemicals, Pearson=Pearson correlation coefficient,

POD=point of departure, RMSE=root mean squared error, RMU=root mean uncertainty, R2=coefficient of determination, Spearman=Spearman correlation coefficient

Figure S1-22. External prediction performance and uncertainty calibration for the non-standardized test set predicted by CP models trained with the standardized dataset for POD_{nc} . **(a)** Prediction performance on the challenging, external test data set of non-standardized chemicals **(b)** The mean prediction uncertainty (RMU) and observed prediction errors (RMSE) for non-standardized chemicals compared to standardized chemicals (during cross-validation). **(c)** Distribution of prediction uncertainty (95% CI width) for non-standardized chemicals compared to standardized chemicals (during cross-validation) **(e)** Confidence-based calibration curve for non-standardized chemicals **(f)** Error-based calibration curve for non-standardized chemicals. **Abbreviations:** ENCE=expected normalized calibration error, CI=confidence interval, CV=cross-validation, Jaccard=mean Jaccard distances from 5 nearest neighbors, MAE=mean absolute error, MdAE=median absolute error, n=number of chemicals, Pearson=Pearson correlation coefficient,

POD=point of departure, RMSE=root mean squared error, RMU=root mean uncertainty, R2=coefficient of determination, Spearman=Spearman correlation coefficient

Comment 4: *The generation of this extensive prediction dataset with uncertainty and PODs hold significant potential value. Are there any plans to make these predictions and their uncertainties known to the wider scientific community or fully public accessible?*

Answer and action: Thank you for recognizing the potential value of this work. Indeed, the predictions for the >100,000 marketed chemicals, along with the trained models, will be made fully and openly available. We will provide complete code and documentation on GitHub (<https://github.com/kejbo/poduam>), as well as a Shiny web application (<https://dtu-quantitative-sustainability-assessment.shinyapps.io/poduam>) that allows users to run the models and explore predictions directly in the browser without requiring local installation. Details are provided in the Data Availability and Code Availability sections.

Comments on Code availability: *The overall quality of the repository and its reproducibility is very good. Several cross-validation scripts have been tested and yielded similar results with the reported findings. Here are some suggestions for implementation and replication,*

- *The Mac OS users need to install CMake to compile code for some dependencies (e.g. dm-tree). The README should add instructions on install build tools.*

- *The README doc lacks instructions and commands on how to run the experiments. Adding sample commands or scripts would improve readability. For example, adding the command `$ python ./manuscript/2-1_performance/run_cv_BNN.py` or `$ python train_final_models.py` would be helpful.*

- *For all training files, the replication could be improved by avoiding hardcoding values for parameter settings. Authors could consider use package like argparse to allow for more convenient replication or experiments, or maybe consider use configuration file (e.g., JSON) for easily editing the key parameters.*

- *The current scripts assume that hyper-parameters are already considered "optimal". The nested cross-validation approach with hyper-parameters optimization has been described in the supplementary section but not mentioned in the repository. The README could provide a navigation to that supplementary section for further implementation details.*

Answer and action: Thank you very much for your detailed review of our code. Based on your suggestions, we have improved the README file to include instructions for Mac users and a clearer overview of the analysis structure, indicating where each part can be found and how to execute the scripts in the correct order. The scripts were developed and tested in an IDE (VS Code), which facilitates adjusting settings at the beginning of each script. To improve the compatibility with command-line execution, we have implemented argument parsing in all scripts where users need to select between different dataset configurations.

Nature communications manuscript NCOMMS-25-15028– revised version

Detailed answers to reviewers' comments:

Reviewer #1

Comment: *The revised manuscript clarified the open points. The manuscript is well-structured and explicative.*

Answer: Thank you for the positive evaluation of our revised work. We are pleased that we successfully addressed all of your comments and concerns.

Reviewer #2

Comment: *The revised manuscript now describes the potential to apply results from the manuscript to inform decisions regarding limitations on use of potentially harmful chemicals and the potential to guide future improvements in predictive performance for chemical classes that are difficult to predict due to high uncertainty. These revisions incompletely address concerns regarding relatively weak novelty and the important potentials described in the revision – though testable – remain theoretical at this stage of investigation. This is important work that when carried forward to realize the above potentials will represent a major scientific contribution.*

Answer and action: Thank you for your thoughtful comments and for recognizing the principal scientific contribution of our work. We agree that further evaluation of the potential to guide reductions in human health impacts and to improve effect data predictions in high-uncertainty chemical domains would provide substantial additional value. However to arrive at risk or health impact level, extensive new data collection efforts would be required – such as chemical production data, data on chemical use in consumer products and industrial processes, indoor/outdoor fate and distribution, and human exposure data to rank health risks from predicted chemicals – as well as generating new experimental data in chemical domains with high uncertainty to improve prediction accuracy in these domains. We consider these important aspects that require substantial additional research effort, while they can already build on and be combined with our effect predictions as an essential component.

To address these concerns, we added a paragraph in the Introduction emphasizing the novel aspects of our work. In the Discussion, we further clarified why demonstrating the potential applications for human health risk and impact assessments was beyond the scope of the present study.

Introduction

L100-105: *“This study provides the first comprehensive demonstration of uncertainty-aware toxicity modeling at a chemical space of this scale, enabling transparent predictions that support confidence-building in machine learning-based chemical assessments. Unlike prior studies, which have focused mainly on predictive accuracy, we quantify and validate prediction uncertainty, addressing a key limitation to the uptake of predicted data in decision-making tools.”*

Discussion

L424-447: “Our models and predicted POD data were developed for application in support of broader chemical impact and risk assessments across marketed chemicals. Such assessments can build on our non-cancer human toxicity predictions as an essential component and combine them with additional information to derive risk or health impact levels, requiring data on chemical production, chemical use in consumer products and industrial processes, indoor and outdoor environmental fate, and human exposure, to quantify chemical-related health risks. For this application, the provided quantitative confidence intervals prevent overconfidence in prediction results and allow verifying if the associated uncertainty can alter or hinder robust conclusions from such chemical assessments. Depending on the decision context, assessments can leverage the best estimates and associated confidence intervals for robust comparative analysis, as in life cycle assessments - or take a more protective approach by considering the upper toxicity bounds to establish safety margins, as is common when assessing human health risks. This is a substantial advancement in departing from the “no data, no problem” mindset toward assessing chemical toxicity potency as needed based on the most likely or conservative estimates while leveraging the same data source. In addition, the uncertainty quantification provided can help prioritize experimental studies to reduce uncertainty in challenging chemical domains, iteratively improving the reliability of risk or health impact level assessments. This is an essential step toward harmonizing data generation for chemical risk and sustainability assessments, addressing potentially conflicting results between these two approaches³⁶ – an increasingly important challenge in the context of integrated frameworks such as the EU Commission’s Safe and Sustainable by Design.¹⁷ While implementing such integrated assessments is beyond the scope of the current study, our well-calibrated predictions and confidence intervals provide a solid foundation for future studies to carry out comprehensive risk and impact evaluations.”

Reviewer #3

Comment: The authors have addressed all my concerns. I appreciate their thorough response and efforts to improve the manuscript. In addition, the code has been significantly improved in clarity and documentation. I recommend this manuscript for publication.

Answer: Thank you for the positive evaluation of our revised work. We are pleased that we successfully addressed all of your comments and concerns.